# Origin of wiring specificity in an olfactory map revealed by neuron type–specific, time-lapse imaging of dendrite targeting

Kenneth Kin Lam Wong[1], Tongchao Li[1]*[†], Tian-Ming Fu[2‡], Gaoxiang Liu[3], Cheng Lyu[1], Sayeh Kohani[1], Qijing Xie[1], David J Luginbuhl[1], Srigokul Upadhyayula[3,4,5], Eric Betzig[2,3,6], Liqun Luo[1]*

[1]Department of Biology, Howard Hughes Medical Institute, Stanford University, Stanford, United States; [2]Howard Hughes Medical Institute, Janelia Research Campus, Ashburn, United States; [3]Advanced Bioimaging Center, Department of Molecular and Cell Biology, University of California, Berkeley, Berkeley, United States; [4]Molecular Biophysics and Integrated Bioimaging Division, Lawrence Berkeley National Laboratory, Berkeley, United States; [5]Chan Zuckerberg Biohub, San Francisco, United States; [6]Departments of Molecular and Cell Biology and Physics, Howard Hughes Medical Institute, Helen Wills Neuroscience Institute, University of California, Berkeley, United States

**\*For correspondence:**
ltongchao@outlook.com (TL);
lluo@stanford.edu (LL)

**Present address:** [†]Liangzhu Laboratory, MOE Frontier Science Center for Brain Science and Brain-machine Integration, State Key Laboratory of Brain-machine Intelligence, Zhejiang University, Hangzhou, China; [‡]Department of Electrical and Computer Engineering, Princeton University, Princeton, United States

**Competing interest:** The authors declare that no competing interests exist.

**Abstract** How does wiring specificity of neural maps emerge during development? Formation of the adult *Drosophila* olfactory glomerular map begins with the patterning of projection neuron (PN) dendrites at the early pupal stage. To better understand the origin of wiring specificity of this map, we created genetic tools to systematically characterize dendrite patterning across development at PN type–specific resolution. We find that PNs use lineage and birth order combinatorially to build the initial dendritic map. Specifically, birth order directs dendrite targeting in rotating and binary manners for PNs of the anterodorsal and lateral lineages, respectively. Two-photon– and adaptive optical lattice light-sheet microscope–based time-lapse imaging reveals that PN dendrites initiate active targeting with direction-dependent branch stabilization on the timescale of seconds. Moreover, PNs that are used in both the larval and adult olfactory circuits prune their larval-specific dendrites and re-extend new dendrites simultaneously to facilitate timely olfactory map organization. Our work highlights the power and necessity of type-specific neuronal access and time-lapse imaging in identifying wiring mechanisms that underlie complex patterns of functional neural maps.

## Editor's evaluation

When a neuron is born it correlates with where it targets in the neuropil and this has been best demonstrated in the olfactory lobe of *Drosophila*. This important study uses sophisticated genetics and advanced live imaging to provide a compelling description of how neuronal dendrites explore the target field, eliminate excessive branches, and assort into the correct region during development. In the process, it develops valuable tools. The study brings us closer to a comprehensive understanding of how the birth order of a neuron translates to dendrite patterning within the *Drosophila* antennal lobe circuit.

**eLife digest** The brain's ability to sense, act and remember relies on the intricate network of connections between neurons. Organization of these connections into neural maps is critical for processing sensory information. For instance, different odors are represented by specific neurons in a part of the brain known as the olfactory bulb, allowing animals to distinguish between smells.

Projection neurons in the olfactory bulb have extensions known as dendrites that receive signals from sensory neurons. Scientists have extensively used the olfactory map in adult fruit flies to study brain wiring because of the specific connections between their sensory and projection neurons. This has led to the discovery of similar wiring strategies in mammals. But how the olfactory map is formed during development is not fully understood.

To investigate, Wong et al. built genetic tools to label specific types of olfactory projection neurons during the pupal stage of fruit fly development. This showed that a group of projection neurons directed their dendrites in a clockwise rotation pattern depending on the order in which they were born: the first-born neuron sent dendrites towards the top right of the antennal lobe (the fruit fly equivalent of the olfactory bulb), while the last-born sent dendrites towards the top left.

Wong et al. also carried out high-resolution time-lapse imaging of live brains grown in the laboratory to determine how dendrites make wiring decisions. This revealed that projection neurons send dendrites in all directions, but preferentially stabilize those that extend in the direction which the neurons eventually target. Also, live imaging showed neurons could remove old dendrites (used in the larvae) and build new ones (to be used in the adult) simultaneously, allowing them to quickly create new circuits.

These experiments demonstrate the value of imaging specific types of neurons to understand the mechanisms that assemble neural maps in the developing brain. Further work could use the genetic tools created by Wong et al. to study how wiring decisions are determined in this and other neural maps by specific genes, potentially yielding insights into neurological disorders associated with wiring defects.

## Introduction

Organization of neuronal connectivity into spatial maps occurs widely in the nervous systems across species (*Luo and Flanagan, 2007*; *Cang and Feldheim, 2013*; *Luo, 2021*). For example, in the retinotopic map of the visual system, nearby neurons in the input field project axons to nearby neurons in the target field (*Cang and Feldheim, 2013*). Such a continuous organization preserves spatial relationships in the visual world. Contrary to retinotopy, the olfactory glomerular map consists of discrete units called glomeruli in which input neurons connect with the cognate output neurons based on neuronal type rather than soma position (*Mombaerts et al., 1996*; *Gao et al., 2000*; *Vosshall et al., 2000*). This discrete map represents a given odor by the combinatorial activation of specific glomeruli. Whereas continuous maps are readily built using gradients of guidance cues (*Cang and Feldheim, 2013*), how glomeruli are placed at specific locations in discrete maps is less clear (*Murthy, 2011*). Understanding the developmental origins of these neural maps is fundamental for deciphering the logic of their functional organization through which information is properly represented and processed.

The adult *Drosophila* olfactory map in the antennal lobe (the equivalent of the vertebrate olfactory bulb) has proven to be a powerful model for studying mechanisms of wiring specificity, thanks to the type-specific connections between the presynaptic olfactory receptor neurons (ORNs) and the cognate postsynaptic projection neurons (PNs). Molecules and mechanisms first identified in this circuit have been found to play similar roles in the wiring of the mammalian brain (e.g. *Hong et al., 2012*; *Berns et al., 2018*; *Pederick et al., 2021*). Assembly of the fly olfactory map begins with dendritic growth and patterning of PNs derived primarily from the anterodorsal (adPNs) and lateral (lPNs) lineages and born with an invariant birth order within each lineage (*Jefferis et al., 2001*; *Jefferis et al., 2004*; *Marin et al., 2005*; *Yu et al., 2010*; *Lin et al., 2012*; *Figure 1A and B*). This patterning creates a prototypic olfactory map, prior to ORN axon innervation, indicative of the PN-autonomous ability to target dendrites into specific regions. However, earlier studies could only unambiguously follow the development of one single PN type – DL1 PNs (*Jefferis et al., 2004*). It remains unclear to date how the prototypic olfactory map is organized and what cellular mechanisms PN dendrites use to achieve

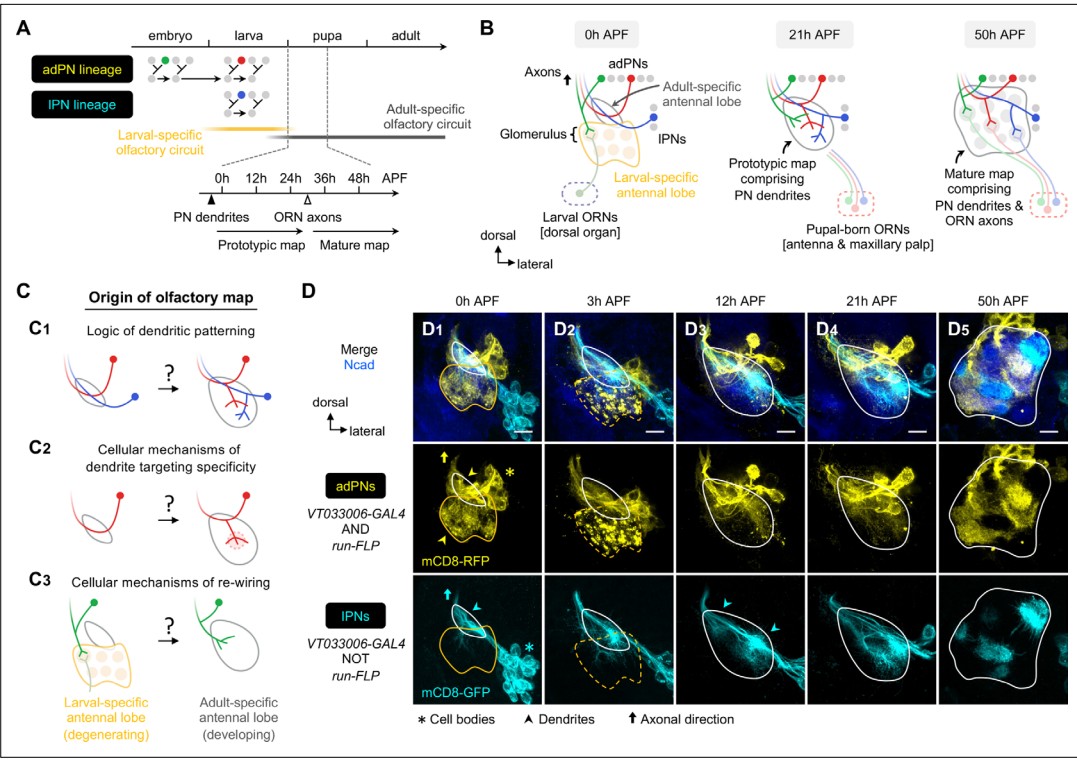

**Figure 1.** Organization and development of the adult olfactory circuit in *Drosophila*. (**A, B**) Timeline (**A**) and schematic illustration (**B**) of *Drosophila* olfactory circuit development. Green, red, and blue circles denote the birth of embryonic-born anterodorsal projection neuron (adPN), larval-born adPN, and larval-born lPN, respectively. At the onset of metamorphosis, the larval-specific olfactory circuit degenerates; larval olfactory receptor neurons (ORNs) die while embryonic-born adPNs prune their larval-specific processes and re-extend new processes into the adult-specific olfactory circuit. In the adult-specific olfactory circuit, projection neuron (PN) dendrites extend first and form a prototypic map. This is followed by an extension of ORN axons and synaptic partner matching between cognate PN dendrites and ORN axons to form a mature map. Solid and open arrowheads in **A** indicate onset of innervation for PN dendrites and ORN axons, respectively. (**C**) Overview of this study investigating the logic of dendritic patterning (**C₁**; see *Figures 3 and 4*) as well as cellular mechanisms of dendrite targeting specificity (**C₂**; see *Figures 6 and 7*) and re-wiring (**C₃**; see *Figure 8*) that contribute to the developmental origin of the adult *Drosophila* olfactory map. (**D**) Staining of fixed brains at indicated stages showing dendrite development of adPNs (*VT033006+ run+*; labeled in yellow) and lPNs (*VT033006+ run–*; labeled in cyan). As *run-FLP* is expressed before 0 h APF in adPN but not lPN neuroblasts, we can use it to label adPNs and lPNs with two distinct colors using an intersectional reporter (see **Materials and methods** for the genotype). Yellow arrowheads in (**D₁**) mark larval- and adult-specific dendrites of adPNs in larval- and adult-specific antennal lobes, respectively. Cyan arrowheads in (**D₃**) denote specific targeting of lPN dendrites at the opposite ends of the dorsomedial-ventrolateral axis. (**D₁**): N=12; (**D₂**): N=7; (**D₃**): N=17; (**D₄**): N=10; (**D₅**): N=12. **Common notations in this study:** Unless otherwise indicated, all images in this and subsequent figures are partial *z* projections of confocal stacks of representative images. *N* indicates the number of antennal lobes imaged. Antennal lobe neuropils are revealed by N-Cadherin (Ncad; in blue) staining. Adult-specific (developing) antennal lobe is outlined with a white solid line. Larval-specific antennal lobe is outlined with an orange line (dashed line used to denote the degeneration stage) and is distinguished from the developing antennal lobe by the more intense nc82 staining as shown in *Figure 1—figure supplement 1* (nc82 channel not shown here). Asterisks (*) indicate PN cell bodies, which are outside the antennal lobe neuropil (and sometimes appear on top because of the z-projections). Arrowheads mark PN dendrites. Arrows mark PN axons projecting towards higher olfactory centers (see *Figure 1—figure supplement 2* for PN axons at their targets in the mushroom body and lateral horn). h APF: hours after puparium formation; h ALH: hours after larval hatching. DL: dorsolateral; DM: dorsomedial; VM: ventromedial; VL: ventrolateral. Scale bar = 10 µm.

The online version of this article includes the following figure supplement(s) for figure 1:

**Figure supplement 1.** Visualization of larval- and adult-specific antennal lobes by co-staining of Ncad and nc82.

**Figure supplement 2.** Projection neuron (PN) axon development across pupal stages.

targeting specificity (*Figure 1C$_{1-2}$*). The initial map formation is further complicated by circuit remodeling during which embryonic-born PNs used in both the larval and adult circuits reorganize their neurites (*Marin et al., 2005*). How embryonic-born PNs coordinate remodeling with re-integration into the adult circuit is not known (*Figure 1C$_3$*).

Here, we set out to explore the origin of the olfactory map by performing a systematic and comparative study of PN dendrite development at type-specific resolution in vivo, and two-photon– and adaptive optical lattice light-sheet microscope–based time-lapse imaging of PN dendrites in early pupal brain explants. As our overarching goal is to understand how the wiring specificity between ORNs and PNs arises, we focus on PNs that project to single glomeruli. Neurons from the lateral lineage that innervate multiple glomeruli or project to other regions of the adult brain (*Lin et al., 2012*) are not studied here. Our study uncovers wiring logic that directs PN dendrites to create an organized olfactory map, dendritic branch dynamics that lead to directional selectivity, and a novel re-wiring mechanism that facilitates timely olfactory map formation. These wiring strategies used in the initial map organization lay the foundation of precise synaptic connectivity between PNs and ORNs in the final glomerular map.

## Results

### Overview of *Drosophila* olfactory circuit development at a lineage-specific resolution

We first described the development of the *Drosophila* olfactory circuit using pupal brains double-labeled for adPNs and lPNs (*Figure 1D*; see the genetic design in *Figure 2*). At the onset of metamorphosis (0 hr after puparium formation; 0 hr APF), the adult-specific antennal lobe (also referred to as 'developing antennal lobe') remained relatively small, located dorsolateral and posterior to the larval-specific antennal lobe (also referred to as 'degenerating antennal lobe') (*Figure 1D$_1$*). As PN dendrites continued to grow and innervate the developing antennal lobe, its size increased considerably (*Figure 1D$_{1-3}$*). By 12 hr APF, PNs already appeared to be sorting their dendrites into specific regions to form a prototypic map, as revealed by the heterogeneous patterning of lPN dendrites (arrowheads in *Figure 1D$_3$*). From 21 hr to 50 hr APF, dendrites of adPNs and lPNs gradually segregated and eventually formed intercalated but non-overlapping glomeruli (*Figure 1D$_{4-5}$*). The development of the adult-specific antennal lobe partially overlapped with the degeneration of the larval-specific antennal lobe, as indicated by fragmentation of the larval-specific dendrites of embryonic-born PNs at 3 hr APF (*Figure 1D$_2$*). This gross characterization at the resolution of two PN lineages was consistent with earlier studies (*Jefferis et al., 2004*; *Marin et al., 2005*). However, the resolution was not sufficiently high to answer the questions we raised in the Introduction (*Figure 1C*).

### Expanded genetic toolkit for type-specific labeling of PNs during early pupal development

To reveal how PN dendrites initiate olfactory map formation at the high spatiotemporal resolution, we needed genetic access to specific PN types during early pupal development. From our recently deciphered single-cell PN transcriptomes (*Xie et al., 2021*), we searched for genetic markers that are expressed strongly and persistently in single or a few PN types across pupal development. This transcriptome-instructed search led to the identification of *CR45223* (in place of this non-coding gene, we used the adjacent *CG14322* that exhibits nearly identical expression pattern), *lov*, and *tsh* (*Figure 2A and B*; *Figure 2—figure supplement 1*).

Next, using CRISPR/Cas9, we generated knock-in transgenic QF2 expression driver lines in which *T2A-QF2* (or *T2A-FLP* for intersection) was inserted immediately before the stop codon of the endogenous gene (*Figure 2—figure supplement 2*). The self-cleaving peptide T2A allows QF2 to be expressed in the same pattern as the endogenous gene (*Diao and White, 2012*). With these new *QF2* lines together with existing *GAL4* lines that label additional PN types (*Xie et al., 2019*), we now have an expanded toolkit accessing PNs ranging from early- to late-born PNs, from adPN to lPN lineages, and from PNs with neighboring glomerular projections to those with distant projections in the adult antennal lobe (*Figure 2C and D*). As QF2/*QUAS* and GAL4/*UAS* expression systems operate orthogonally to each other (*Potter et al., 2010*; *Riabinina et al., 2015*), we crossed our *QF2* lines with existing *GAL4* lines for simultaneous labeling of distinct PN types in the same brain (see inset in *Figure 2C*).

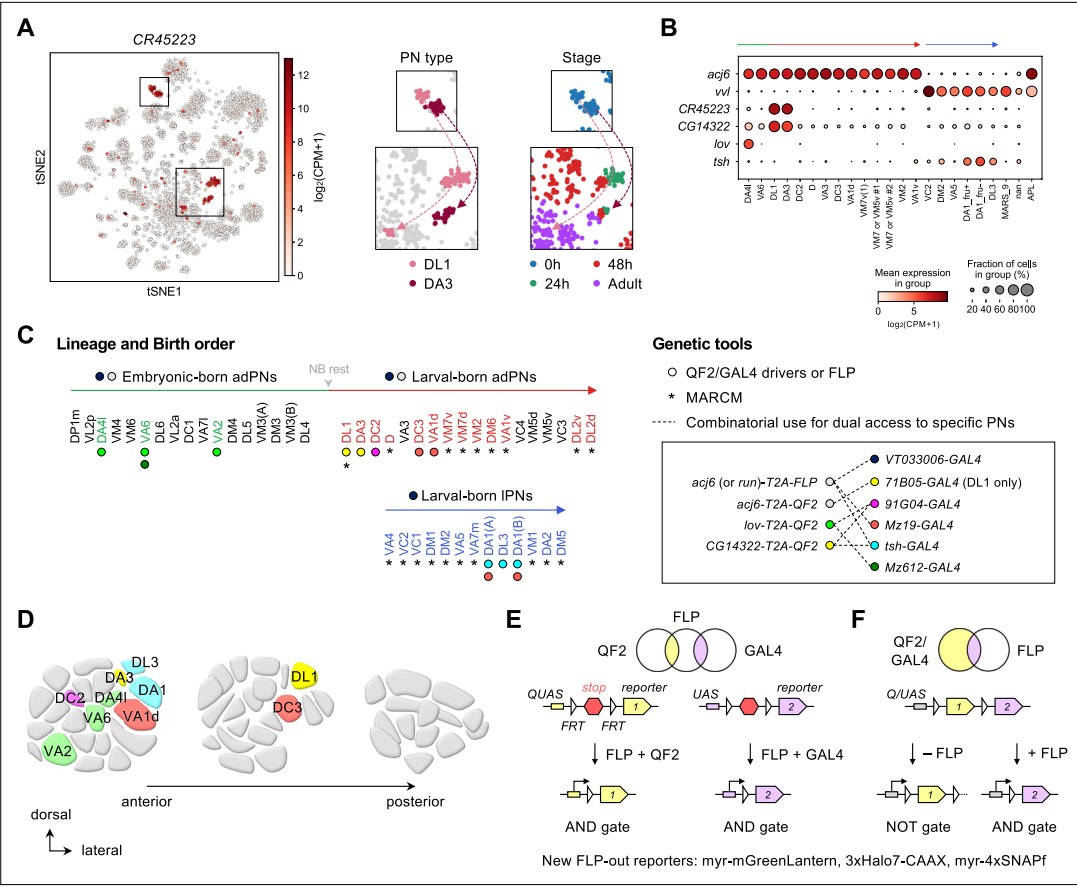

**Figure 2.** Expanded genetic toolkit for dual-color, type-specific labeling of projection neurons (PNs). (**A**) tSNE plot of PN single-cell transcriptomes, color-coded according to *CR45223* expression level in [$\log_2$(CPM +1)], where CPM stands for transcript counts per million reads. Zoom-in of boxes in the tSNE plot (left) is shown on the right, and color-coded according to PN types and developmental stages. (**B**) Dot plot showing the expression of *acj6*, *vvl*, *CR45223*, *CG14322*, *lov*, and *tsh* in 0 hr APF PNs arranged according to their birth order and lineage (green: embryonic-born anterodorsal projection neuron (adPNs); red: larval-born adPNs; blue: larval-born lPNs). Unit of expression is [$\log_2$(CPM +1)] as in **A**. Data from panels A are B are from *Xie et al., 2021*. (**C**) Birth orders of adPNs and lPNs summarized by *Lin et al., 2012*; *Yu et al., 2010* and genetic tools used to access them. **Left:** Accessible PN types are colored. Circles beneath the PN types denote *QF2/GAL4* drivers used to access them. Asterisks beneath the PN types denote access by MARCM. Gray arrowhead marks neuroblast (NB) rest. **Right:** Genetic tools. Inset shows the combinatorial use of *QF2/FLP* and *GAL4* (linked by dashed lines) for comparative analyses of dendrite development of two groups of PNs in the same animal. (**D**) Schematic of glomerular projections of *QF2/GAL4*-accessible PNs in the adult antennal lobe. Indicated glomeruli are color-coded based on the genetic tools used to access them. See the color code in **C**. (**E, F**) Schematic of intersectional logic gates for dual-color labeling of PNs. See *Figure 2—figure supplement 2* for newly generated FLP-out reporters.

The online version of this article includes the following figure supplement(s) for figure 2:

**Figure supplement 1.** Expression of projection neuron (PN) marker genes across development.

**Figure supplement 2.** Generation of *T2A-QF2/FLP* transgenic flies by CRISPR/Cas9.

**Figure supplement 3.** Design of single- and dual-color FLP-out reporters.

This combinatorial use of driver lines permitted comparative analyses of the development of distinct PN types with minimal biological and technical variations (*Supplementary file 1*).

To limit driver expression only in PNs, we applied intersectional logic gates (AND and NOT gates) using our newly generated conditional reporters genetically encoding either mGreenLantern, Halo tags, and/or SNAP tags (*Kohl et al., 2014*; *Sutcliffe et al., 2017*; *Campbell et al., 2020*; *Figure 2E and F*; *Figure 2—figure supplement 3*). These reporters can be broadly used in other systems. Finally,

we used MARCM (*Lee and Luo, 1999*) to label PNs that remain inaccessible due to a lack of drivers (*Figure 2C*; discussed in *Figure 3*).

## Early larval-born adPN dendrites initially share similar targeting regions

Using the new genetic tools, we first re-visited the dendrite development of DL1 PNs—the first larval-born adPN type—using pupal brains double-labeled for DL1 PNs (labeled by *71B05-GAL4*) and adPNs (*Figure 3A*). Consistent with our previous study (*Jefferis et al., 2004*), DL1 PNs already showed robust dendritic growth at the wandering third instar larval stage (*Figure 3—figure supplement 1A*). At 0 hr APF, DL1 PN dendrites extended radially outwards from the main process, reaching nearly the entire developing antennal lobe and often overshooting it (white arrowheads in *Figure 3A$_1$*), likely surveying the surroundings. By 6 hr APF, most of the dendrites already occupied the dorsolateral (DL) corner of the antennal lobe (*Figure 3A$_2$*). As the antennal lobe continued to grow, this dorsolateral positioning of the DL1 PN dendrites remained largely unchanged (*Figure 3A$_{3-6}$*). From 21 hr APF onwards, the dendrites underwent progressive refinement: they were restricted into a smaller area by 30 hr APF (*Figure 3A$_{4-5}$*), and eventually formed a compact, posterior glomerulus by 50 hr APF (*Figure 3A$_6$* showing a single *z* section).

To assess whether other PN types follow the same developmental trajectory, we next examined *CG14322+* PNs, which include DL1 PNs and DA3 PNs—the first and second larval-born adPN types, respectively. In the same brain, we also labeled with a different fluorophore DC2 PNs—the third larval-born adPN type (*Figure 3B*). The dendritic pattern of DL1/DA3 PNs appeared indistinguishable from that of DL1 PNs from 0 hr to 12 hr APF (compare the yellow channel of *Figure 3B$_{1-3}$* with *Figure 3A$_{1-3}$*), suggesting that DL1 and DA3 PN sent dendrites to the same region in the antennal lobe. We began to see differences in 21 hr APF pupal brains in which DL1/DA3 PN dendrites not only occupied the dorsolateral region but also spread ventrally (white arrowhead in *Figure 3B$_4$*; compare with *Figure 3A$_4$*). The more ventrally targeted dendrites likely belong to DA3 PNs. This suggests that ~21 hr APF marks the beginning of dendritic segregation of DL1 and DA3 PNs. By 30 h APF, DL1 and DA3 dendrites were clearly separable (*Figure 3B$_5$*), which respectively formed more posteriorly and anteriorly targeted glomeruli at 50 hr APF (*Figure 3B$_6$*; see single *z* sections in *Figure 3—figure supplement 1C*).

Next, we focused on the third-born—DC2 PNs labeled by *91G04-GAL4* (*Figure 3B*). This *GAL4* labeled additional embryonic-born adPNs from 0 hr to 6 hr APF, but the expression in these PNs diminished afterward. As embryonic-born adPNs do not have any dendrites in the developing antennal lobe at 0 hr APF (discussed in Figure 8), dendrites found in the antennal lobe should belong to the larval-born DC2 PNs. Like DL1/DA3 PNs, DC2 PNs initiated radial dendritic extension across the antennal lobe at 0 hr APF (*Figure 3B$_1$*; *Figure 3—figure supplement 1B*). Notably, DL1/DA3 and DC2 PN dendrites exhibited substantial overlap from 0 hr to 12 hr APF and shared a similar targeting region at the dorsolateral corner from 6 hr to 12 hr APF (*Figure 3B$_{1-3}$*). It was not until 21 hr APF that DL1, DA3, and DC2 dendrites began to segregate from each other along both medial-lateral and anterior-posterior axes (*Figure 3B$_{4-5}$*). By 50 hr APF, the DC2 glomerulus was separated from DL1/DA3 glomeruli by intermediate glomeruli (*Figure 3B$_6$*).

In summary, dendrites of consecutively larval-born DL1, DA3, and DC2 adPNs (here collectively named 'early larval-born adPNs'; see its definition in next section) develop in a similar fashion and share a similar targeting region at early pupal stages (0–12 hr APF). This is then followed by their segregation into distinct regions close to their adult glomerular positions during mid-pupal stages (21–50 hr APF).

## Larval-born adPNs with distant birth order send dendrites to distinct regions

The analysis of early larval-born adPNs (*Figure 3A and B*) led us to hypothesize that larval-born adPNs might use their birth order to coordinate dendrite targeting during early pupal stages. If this were true, we would expect dendrites of larval-born adPNs with distant birth order to occupy distinct regions. To test this hypothesis, we compared dendrite-targeting regions of early larval-born adPNs with those of later-born adPNs.

We first examined DC3/VA1d adPNs (referred to as 'mid-early larval-born adPNs') using *Mz19-GAL4* (*Figure 3C*). This *GAL4* is expressed in three PN types from 24 hr APF to adulthood: DC3 adPNs, VA1d

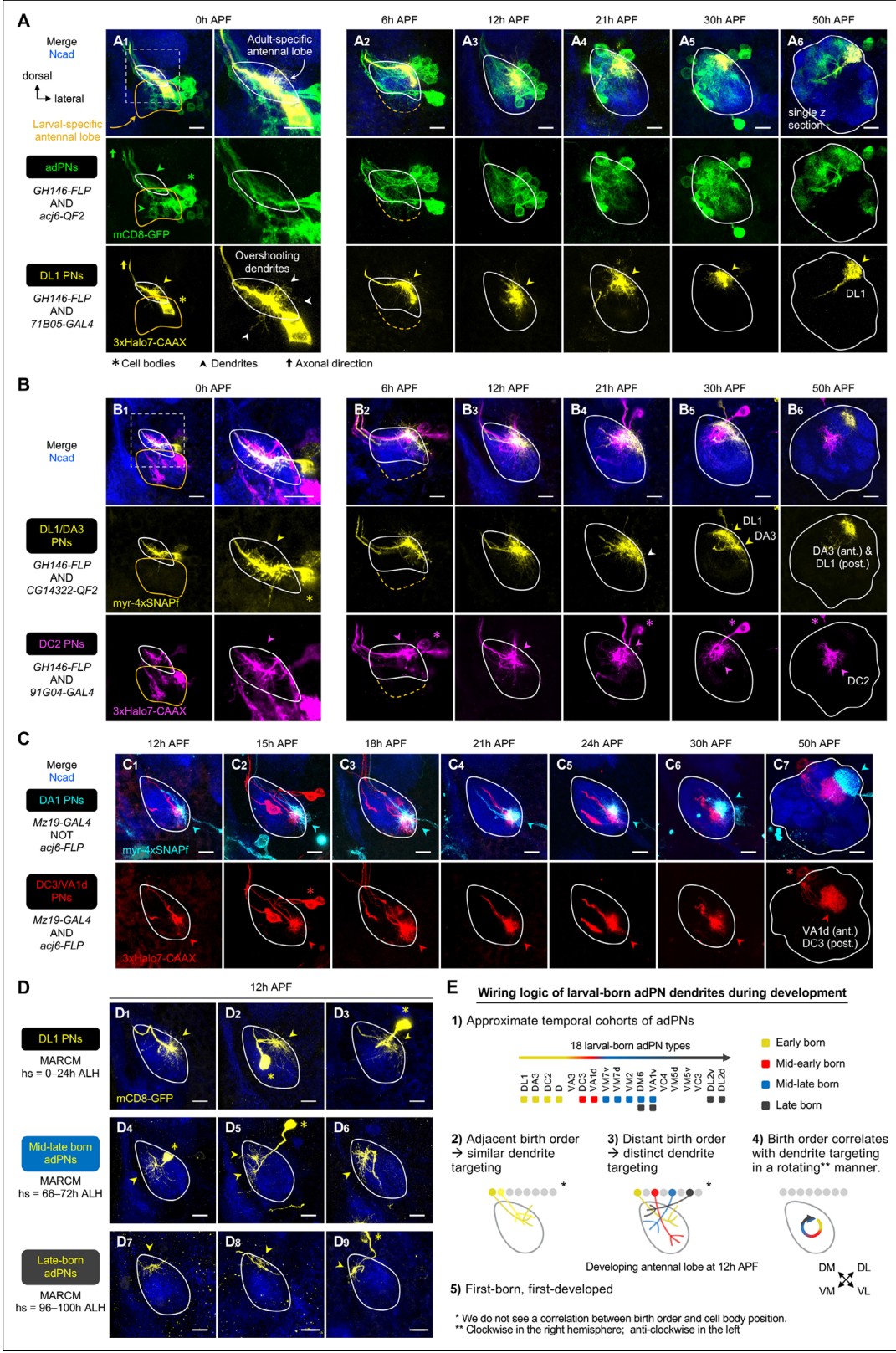

**Figure 3.** Birth order–dependent spatial patterning of anterodorsal projection neuron (adPN) dendrites in the developing antennal lobe. (**A**) Confocal images of fixed brains at indicated stages showing dendrite development of adPNs (*acj6+*; labeled in green) and DL1 adPNs (*71B05+*; labeled in yellow). Right column of A₁ shows a zoom-in of the dashed box. The labeling of *acj6+* adPNs outlines the developing antennal lobe and is used in dual-color

*Figure 3 continued on next page*

*Figure 3 continued*

AO-LLSM imaging later (see *Figure 7A–C*). White arrowheads in ($A_1$) mark dendrites overshooting the antennal lobe. ($A_1$): N=14; ($A_2$): N=12; ($A_3$): N=14; ($A_4$): N=6; ($A_5$): N=4; ($A_6$): N=4. (**B**) Confocal images of fixed brains at indicated stages showing dendrite development of DL1/DA3 adPNs (*CG14322+*; labeled in yellow) and DC2 adPNs (*91G04+*; labeled in magenta). As *91G04-GAL4* labels some embryonic-born projection neurons (PNs) from 0 to 6 hr APF, their neurites are found in the larval-specific antennal lobe ($B_{1,2}$). Right column of ($B_1$) shows a zoom-in of the dashed box. White arrowhead in ($B_4$) denotes the more ventrally targeted DL1/DA3 dendrites. ($B_1$): N=6; ($B_2$): N=5; ($B_3$): N=12; ($B_4$): N=4; ($B_5$): N=7; ($B_6$): N=2. (**C**) Confocal images of fixed brains at indicated stages showing dendrite development of DC3/VA1d adPNs (*Mz19+ acj6+*; labeled in red) and DA1 lPNs (*Mz19+ acj6–*; labeled in cyan). ($C_1$): N=14; ($C_2$): N=6; ($C_3$): N=4; ($C_4$): N=10; ($C_5$): N=10; ($C_6$): N=6; ($C_7$): N=4. (**D**) Confocal images of single-cell MARCM clones (in yellow) of DL1 PNs ($D_{1-3}$), mid-late larval-born adPNs ($D_{4-6}$), and late larval-born adPNs ($D_{7-9}$) in 12 hr APF pupal brains, generated by heat shocks (hs) at indicated times. Three biological samples are shown for each of the indicated adPN cohorts. $D_{1-3}$: N=5; $D_{4-6}$: N=4; $D_{7-9}$: N=8. (**E**) Summary of wiring logic of larval-born adPN dendrites to form an olfactory map in the 12 hr APF developing antennal lobe. See *Figure 1* legend for common notations.

The online version of this article includes the following video, source data, and figure supplement(s) for figure 3:

**Figure supplement 1.** Dendrite development of early larval-born projection neurons (PNs).

**Figure supplement 2.** MARCM-labeled single-cell projection neurons (PNs) of indicated lineages in adult brains.

**Figure supplement 3.** Dendrite development of DL1, middle larval-born, and late larval-born projection neurons (PNs) at early stages.

**Figure supplement 3—source data 1.** Source data for *Figure 3—figure supplement 3F and G*.

**Figure 3—video 1.** 3D rendering of *z* stacks of indicated projection neurons (PNs) in 12 hr APF antennal lobe. https://elifesciences.org/articles/85521/figures#fig3video1

---

adPNs, and DA1 lPNs (*Jefferis et al., 2004*). To distinguish adPNs from lPNs, we previously adopted an FLP-out strategy labeling *Mz19+* PNs with either GFP or RFP based on their lineages and studied dendrite segregation and refinement during mid-pupal stages (*Li et al., 2021*; *Figure 3C$_{4-7}$*). However, the weak *GAL4* expression before 24 hr APF prevented us from visualizing any dendrites at earlier stages. To overcome this, we incorporated Halo and SNAP chemical labeling (*Kohl et al., 2014*) in place of the immunofluorescence approach. This modification substantially extended the detection to developmental stages as early as 12 hr APF (*Figure 3C$_1$*). We found that, from 12 hr to 21 hr APF, DC3/VA1d PN dendrites targeted the ventrolateral (VL) corner of the antennal lobe (*Figure 3C$_{1-4}$*). Thus, early (DL1/DA3/DC2) and mid-early (DC3/VA1d) larval-born adPN dendrites occupy distinct regions at 12 hr APF.

As we did not have reliable drivers to access other later-born PNs at early pupal stages, we turned to MARCM (*Lee and Luo, 1999*) to generate heat shock-induced single-cell clones of PNs born at different times (*Figure 3—figure supplement 2*). We used *GH146-GAL4(IV)*, a PN driver that labels the majority of PN types, including later-born adPNs (*Figure 3—figure supplement 2D–E*), with a tight temporal control of heat shock and analyzed heat shock-induced animals that were among the first to form puparium to minimize the effects of unsynchronized development among individual animals (see **Materials and methods** for details). These optimizations permitted a systematic clonal analysis at higher PN type-specific resolution that correlates with birth time.

Based on birth timing that corresponds to the heat shock time we applied to induce single-cell MARCM clones, we assigned larval-born adPNs to approximate temporal cohorts: (1) heat shock at 0–24 hr ALH (after larval hatching): first-born (DL1), (2) heat shock at 42–48 hr ALH: early-born (DL1, DA3, DC2, and D), (3) heat shock at 66–72 hr ALH: mid-late born (VM7v, VM7d, VM2, DM6, and VA1v), and (4) heat shock at 96–100 hr ALH: late-born (DM6, VA1v, DL2v, DL2d) (*Figure 3E$_1$*). We assigned DC3/VA1d PNs labeled by *Mz19-GAL4* to the mid-early cohort because they are born between the early and mid-late adPNs. We note that DM6 and VA1v PNs were assigned to both cohorts of mid-late and late-born adPNs, reflecting the nature of short birth timing differences and overlaps between adjacent cohorts. Using this strategy, we could also label lPNs born at different times and assigned them into approximate temporal cohorts (*Figure 3—figure supplement 2F*).

Clonal analysis revealed that, at 12 hr APF, the first-born DL1 adPNs sent dendrites to the dorso-lateral corner of the antennal lobe as expected (*Figure 3D$_{1-3}$*). By contrast, dendrites of mid-late larval-born adPNs occupied a large region on the medial/dorsomedial (M/DM) side (*Figure 3D$_{4-6}$*).

The dendritic arborization patterns of these PNs varied widely, most likely because they belonged to different PN types. Intriguingly, late larval-born adPN dendrites targeted the peripheral, dorsomedial (abbreviated as pDM) corner where the staining of the pan-neuropil marker N-Cadherin was relatively weak (*Figure 3D7–9*). The weak staining implies that this area is less populated by PN dendrites (the major constituent of the antennal lobe neuropil at this stage), possibly because (1) this area is not innervated by many PNs and/or (2) the dendrites of late-born PNs innervate later and remain less elaborate than earlier-born PNs (we will explore this later).

Together, our data (*Figure 3A–D*) suggest that larval-born adPNs with adjacent birth order send dendrites to similar regions of the developing antennal lobe whereas those with distant birth order send dendrites to distinct regions (*Figure 3E2,3*). Notably, the birth order of the examined PNs does not specify dendrite targeting randomly (*Figure 3E4*). Rather, the stereotyped dendritic pattern in the prototypic map correlates with the birth order in an organized manner (rotating clockwise in the right hemisphere when viewed from the front; anti-clockwise in the left: early↔DL; mid-early↔VL; mid-late↔M/DM; late↔pDM). One can, therefore, infer at least the approximate birth order of a larval-born adPN based on its initial dendrite targeting, and *vice versa*.

As the antennal lobe is a 3D structure, we also visualized PN dendrite targeting in the 12 hr APF map with 3D rendering generated from *z* stacks with rotation along the *y*-axis (*Figure 3—video 1*). We found that, along the short anterior-posterior axis (spanning about 20 μm), PN dendrites were located primarily on the periphery of the antennal lobe, whereas the center housed the axon bundle projecting out of the antennal lobe. Some dendrites could reach almost the entire depth, suggesting active exploration of the surroundings in many directions. While 3D projections provide rich details in depth and different viewing angles, we did not find an apparent relationship between birth order and dendrite targeting along the anterior-posterior axis, at least for the examined PN types at 12 hr APF. Thus, the approximate 2D projection (*Figure 3E2–4*) conveys the logic of dendrite patterning effectively.

## Dendrite targeting timing of larval-born adPN depends on birth order

Having provided evidence for birth order–dependent spatial patterning of larval-born adPN dendrites, we next asked whether the timing of dendritic extension and targeting is also influenced by birth order. We noticed that the extent of dendritic innervation of 0 hr APF first-born DL1 adPNs resembled that of 6 hr APF mid-late born adPNs (compare *Figure 3—figure supplement 3A1–4* with *Figure 3—figure supplement 3B5–8*). Such a resemblance was also seen between 0 hr APF mid-late and 6 hr APF late-born adPNs (compare *Figure 3—figure supplement 3B1–4* with *Figure 3—figure supplement 3C*). Quantitative analyses of the exploring volume of dendrites and the number of terminal branches showed that, at 0 hr APF, DL1 PN dendrites were more elaborate than mid-late born PN dendrites (*Figure 3—figure supplement 3F*). By 6 hr APF, the mid-late born appeared to catch up, showing an extent of innervation comparable to DL1 PNs.

We next examined when the dendrites reach their targeting regions. We found that whereas early larval-born adPNs (DL1, DA3, DC2) concentrated their dendrites to the dorsolateral corner by 6 hr APF (*Figure 3B2*; *Figure 3—figure supplement 3A5–8*), later-born PNs concentrated their dendrites to the medial/dorsomedial or peripheral dorsomedial side at 12 hr APF (*Figure 3D4-9*; *Figure 3—figure supplement 3B5-8, C*). Thus, our results suggest larval-born adPN dendrites innervate and pattern the antennal lobe using a 'first born, first developed' strategy.

## Contribution of lineage to early PN dendritic patterning

Both lineage and birth order of PNs contributes to the eventual glomerular choice of their dendrites (*Jefferis et al., 2001*). What is the involvement of lineage in the prototypic map formation? Do lPN dendrites pattern the developing antennal lobe following similar rules as adPNs? To characterize lPN dendrite development at type–specific resolution, we used *tsh-GAL4* to genetically access DA1/DL3 lPNs, and MARCM clones of lPNs as a complementary approach (*Figure 4*). We focused on the dendritic patterns of *tsh+* DA1/DL3 lPNs from 0 hr to 12 hr APF as *tsh-GAL4* labeled additional PNs from 21 hr APF onwards (*Figure 4A4–6*; *Figure 4—figure supplement 1B4–6*; *Figure 4—figure supplement 2*; *Figure 2—figure supplement 1*).

Examination of pupal brains double-labeled with DA1/DL3 lPNs (referred to as 'middle larval-born lPNs') and DL1/DA3 adPNs revealed that, like the early larval-born adPNs, dendritic growth of DA1/

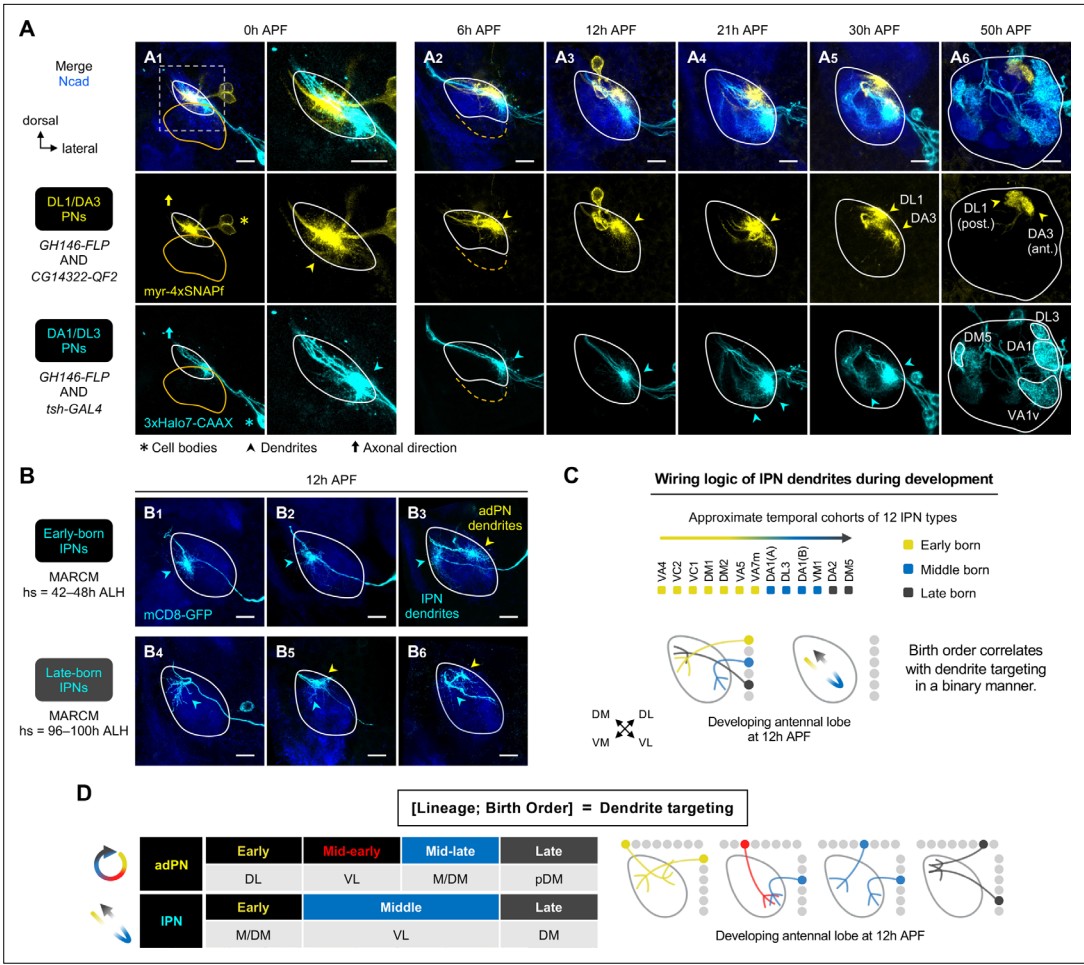

**Figure 4.** Birth order–dependent spatial patterning of lPN dendrites in the developing antennal lobe. (**A**) Confocal images of fixed brains at indicated stages showing dendrite development of DL1/DA3 adPNs (*CG14322+*; labeled in yellow) and DA1/DL3 lPNs (*tsh+*; labeled in cyan). Right column of $A_1$ shows a zoom-in of the dashed box. (**A₁**): N=8; (**A₂**): N=4; (**A₃**): N=6; (**A₄**): N=10; (**A₅**): N=4; (**A₆**): N=5. (**B**) MARCM clones (in cyan) of early (**B₁₋₃**) and late (**B₄₋₆**) larval-born lPNs in 12 hr APF pupal brains, generated by heat shocks (hs) at indicated times. In (**B₃**), (**B₅**), and (**B₆**), single-cell clones of anterodorsal projection neuron (adPN) (yellow arrowheads) and lPN (cyan arrowheads) lineages were simultaneously labeled. Three biological samples are shown for each of the indicated lPN cohorts. **B₁₋₃**: N=4; **B₄₋₆**: N=6. (**C**) Summary of wiring logic of larval-born lPN dendrites to form an olfactory map in the 12 hr APF developing antennal lobe. (**D**) Summary of determination of dendrite targeting of larval-born PNs by lineage and birth order. See *Figure 1* legend for common notations.

The online version of this article includes the following video and figure supplement(s) for figure 4:

**Figure supplement 1.** Dendrite development of DL1/DA3 and DA1/DL3 projection neurons (PNs).

**Figure supplement 2.** Expression patterns of *tsh* in the developing antennal lobe during mid-pupal stages.

**Figure 4—video 1.** 3D rendering of *z* stacks of indicated projection neurons (PNs) in 12 hr APF antennal lobe.

https://elifesciences.org/articles/85521/figures#fig4video1

---

DL3 lPNs was evident by the wandering third instar larval stage (*Figure 4—figure supplement 1A*). At this stage, most DA1/DL3 lPN dendrites innervated the antennal lobe and intermingled with those of DL1/DA3 adPNs. From 0 hr to 12 hr APF, despite a high degree of overlap among those dendrites that explored the surroundings, DA1/DL3 lPN dendrites primarily targeted an area ventrolateral to those of DL1/DA3 adPNs (*Figure 4A₁₋₃*; see 3D rendering in *Figure 4—video 1*). Such a spatial distinction was also observed between middle larval-born adPNs and lPNs in 0 hr and 6 hr APF pupal brains where occasionally single-cell clones from both lineages were simultaneously generated by MARCM

(*Figure 3—figure supplement 3D$_{1–4, 7–10}$*). Thus, at least some adPNs and lPNs sort their dendrites into distinct regions very early on regardless of birth timing.

Next, we used MARCM to ask if lPNs born earlier and later than DA1/DL3 lPNs would send dendrites to regions different from that of DA1/DL3 lPNs. We found that dendrites of early-born lPNs primarily occupied the medial/dorsomedial side of the antennal lobe (*Figure 4B$_{1–3}$*); we note that adPNs born at the same time sent dendrites to the dorsolateral side (see yellow arrowhead in *Figure 4B$_3$*). Also, in contrast to the ventrolateral targeting of middle-born lPN dendrites, late-born lPNs sent dendrites to the dorsomedial corner (*Figures 4B$_{4–6}$*). Like larval-born adPNs, late-born lPNs innervated the antennal lobe later than earlier-born lPNs (*Figure 3—figure supplement 3D$_{7–12}$–E, G*).

These data suggest that, at early pupal stages, lPN dendrites pattern the developing antennal lobe following similar rules as larval-born adPNs: adjacent birth order → similar dendrite targeting; distant birth order → distinct dendrite targeting; 'first born, first developed.' However, unlike the correlation of birth order and target positions in a rotational manner for adPNs (*Figure 3E*), the lPN dendritic map formation appears binary: early↔M/DM; middle↔VL; late↔DM (*Figure 4C*). Our type-specific characterization corroborated with the gross examination of the lPN dendrites as previously reported (*Jefferis et al., 2004*): at 12 hr APF, lPN dendrites mostly occupied the opposite corners along the dorsomedial-ventrolateral axis, leaving the middle of the axis largely devoid of lPN dendrites (arrowheads in *Figure 1D$_3$*).

In summary, we propose that lineage and birth order of larval-born PNs contribute to their dendrite targeting in a combinatorial fashion (*Figure 4D*). The wiring logic of PN dendrites in the developing antennal lobe can, therefore, be represented by [lineage, birth order]=dendrite targeting; one can deduce the unknown if the other two are known.

## An explant system for time-lapse imaging of PN development at early pupal stages

So far, we have identified wiring logic governing the initial dendritic map formation (*Figures 3 and 4*) by examining specifically labeled neuron types in the fixed brain at different developmental stages. To examine dendrite targeting at the higher spatiotemporal resolution, we established an early-pupal brain explant culture system based on previous protocols (*Özel et al., 2015*; *Rabinovich et al., 2015*; *Li and Luo, 2021*; *Li et al., 2021*), and performed single- or dual-color time-lapse imaging with two-photon microscopy as well as adaptive optical lattice light-sheet microscopy (AO-LLSM) (*Figure 5A–C*). The following lines of evidence support that our explant system recapitulates key features of in vivo olfactory circuit development.

First, during normal development, the morphology of the brain lobes changes from spherical at 0 hr APF to more elongated rectangular shapes at 15 hr APF (*Rabinovich et al., 2015*). After 22 hr ex vivo culture, the spherical hemispheres of brains dissected at 3 hr APF became more elongated, mimicking ~15 hr APF in vivo brains characterized by the separation of the optic lobes from the central brain (*Figure 5D*).

Second, dual-color, two-photon imaging of PNs every 20 min for 22 hr revealed that lPNs in 3 hr APF brains initially produced dynamic but transient dendritic protrusions in many directions, followed by extensive innervation into the antennal lobe (arrowheads in *Figure 5E$_{1–3}$*; *Figure 5—video 1*). In higher brain centers, lPN axons clearly showed direction-specific outgrowth of collateral branches into the mushroom body calyx as well as forward extension into the lateral horn (arrows in *Figure 5E$_3$*), thus resembling in vivo development (*Figure 1—figure supplement 2*).

Third, larval-specific dendrites observed in 0 hr APF brains cultured for 12 hr ex vivo (orange arrowhead in *Figure 5F$_4$*) were no longer seen in those cultured for 24 hr ex vivo (*Figure 5F$_5$*), indicative of successful pruning and clearance of larval-specific dendrites. Also, the size of the developing antennal lobe in the brains cultured for 24 hr ex vivo increased considerably (*Figure 5F$_5$*). These imply that olfactory circuit remodeling (degeneration of larval-specific processes and growth of adult-specific processes) proceeds normally, albeit at a slower rate (compare with *Figure 5F$_{1–3}$*).

Fourth, dendrites from genetically identified DL1 and DA1/DL3 PNs targeted to their stereotyped locations in the antennal lobe in 0 hr APF brains cultured for 24 hr ex vivo (*Figure 5G*), mimicking in vivo development (*Figure 4A*).

Finally, the segregation of dendrites of PNs targeting to neighboring proto-glomeruli could be recapitulated in brains dissected at 24 hr APF and cultured for 8 hr (*Figure 5—figure supplement 1*;

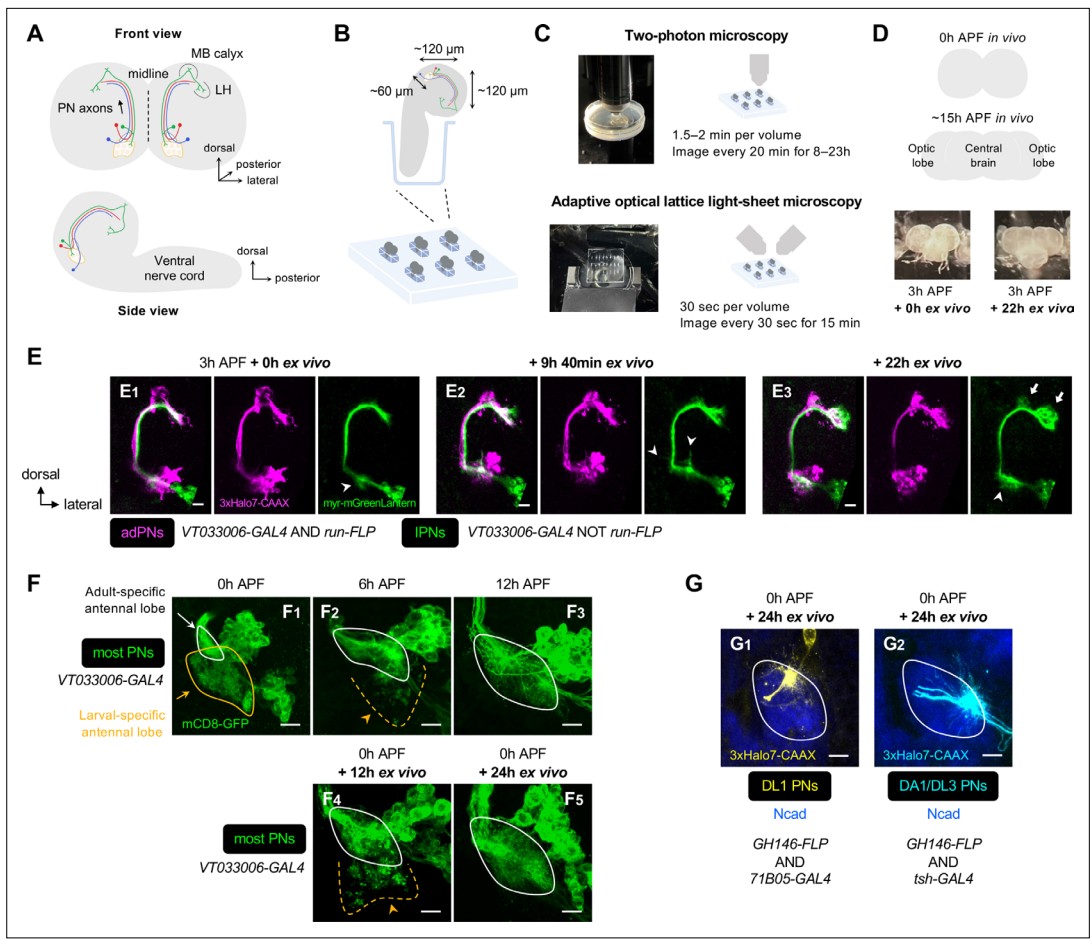

**Figure 5.** Establishment of an explant system for time-lapse imaging of olfactory map formation. (**A**) Schematic of the anatomical organization of the olfactory circuit in early pupal brain (0–3 hr APF). Green, red, and blue denote embryonic-born adPN, larval-born anterodorsal projection neuron (adPN), and larval-born lPN, respectively. MB: mushroom body; LH: lateral horn. (**B**) Schematic of explant culture system for early pupal brains. Wells created in the Sylgard plate from which brains were imbedded are shown in blue. (**C**) Schematic of explant culture and imaging system for early pupal brains. (**D**) **Top:** Schematic of morphological changes of brain lobes from 0 hr to ~15 hr APF during normal development. **Bottom:** Morphologies of a brain explant dissected at 3 hr APF and cultured for 0 hr ex vivo and cultured for 22 hr ex vivo. (**E**) Two-photon time-lapse imaging of adPNs (*VT033006+ run+* ; labeled in magenta) and lPNs (*VT033006+ run–*; labeled in green) in pupal brain dissected at 3 hr APF and cultured for 0–22 hr ex vivo. Arrowheads mark dynamic but transient dendritic protrusions of lPNs in **E**$_{1, 2}$, and extensive dendritic innervation of lPNs in (**E**$_3$). Arrows in (**E**$_3$) mark axonal innervation of lPNs in the mushroom body calyx and lateral horn. N=3. (**F**) Confocal images of antennal lobes labeled by *VT033006+* projection neurons (PNs) (in green) at 0 hr (**F**$_1$), 6 hr (**F**$_2$), and 12 hr (**F**$_3$) APF in vivo. Confocal images of antennal lobes labeled by *VT033006+* PNs in pupal brains were dissected at 0 hr APF and cultured for 12 hr (**F**$_4$) and 24 hr (**F**$_5$) ex vivo. (**F**$_1$): N=6; (**F**$_2$): N=5; (**F**$_3$): N=6; (**F**$_4$): N=8; (**F**$_5$): N=8. (**G**) Dendrite targeting regions of DL1 PNs (*71B05+*; in yellow; **G**$_1$) and DA1/DL3 PNs (*tsh+*; in cyan; **G**$_2$) in the antennal lobes in pupal brains dissected at 0 hr APF and cultured for 24 hr ex vivo. Antennal lobes are revealed by N-Cadherin (Ncad; in blue) staining. (**G**$_1$): N=5; (**G**$_2$): N=6. See **Figure 1** legend for common notations.

The online version of this article includes the following video, source data, and figure supplement(s) for figure 5:

**Figure supplement 1.** Dendritic segregation of DC3/VA1d adPNs and DA1 lPNs targeting neighboring proto-glomeruli.

**Figure supplement 1—source data 1.** Source data for *Figure 5—figure supplement 1C and D*.

**Figure 5—video 1.** Two-photon time-lapse imaging of projection neuron (PN) development.
https://elifesciences.org/articles/85521/figures#fig5video1

**Figure 5—video 2.** Two-photon time-lapse imaging of projection neuron (PN) dendritic segregation.
https://elifesciences.org/articles/85521/figures#fig5video2

*Figure 5—video 2*). Specifically, despite constant dynamic interactions among dendrites that explore the surroundings (arrowheads in *Figure 5—figure supplement 1A$_{2–4}$*), DC3/VA1d and DA1 PNs exhibited a 1–2 µm increase in the distance between centers of the two dendritic masses and a substantial decrease in the overlap of their core targeting regions (*Figure 5—figure supplement 1B–D*). Taken together, these data support that the explant culture and imaging system established here reliably captures key neurodevelopmental events starting from early pupal stages.

## Single-cell, two-photon imaging reveals active dendrite targeting

Our observation in fixed brains revealed that dendrites of DL1 adPNs transition from a uniform extension in the antennal lobe at 0 hr APF to concentration at the dorsolateral corner of the antennal lobe at 6 hr APF (*Figure 3A*). To identify mechanisms of dendrite targeting specificity that could be missed in static developmental snapshots, we performed two-photon time-lapse imaging of single-cell MARCM clones of DL1 PNs in 3 hr APF brains (*Figure 6*; *Figure 6—figure supplement 1*; *Figure 6—video 1*). Although we did not have a counterstain outlining the antennal lobe, we could use the background signals to discern the orientation of DL1 PNs in the brain (*Figure 6—figure supplement 1A*). The final targeting regions relative to the antennal lobe revealed by *post hoc* fixation and immunostaining confirmed proper dendrite targeting (yellow arrowhead in *Figure 6A$_{10}$*; *Figure 6—figure supplement 1B–C*).

Using DL1 PN in *Figure 6A* (pseudo-colored in yellow; *Figure 6—video 1*) as an example, we observed that the PN initially extended dendrites in every direction (*Figure 6A$_{1–3}$*), like what we observed in fixed tissues (*Figure 3A$_1$*). The first sign of active targeting emerged at 2 hr 20 min ex vivo when DL1 PN began to generate long, albeit transient, dendritic protrusions in the dorsolateral direction; these selective protrusions were more prominent at 3 hr ex vivo (arrowheads in *Figure 6A$_{4–6}$*). The dorsolateral targeting continued to intensify, leading to the formation of a highly focal dendritic mass seen at 13 hr ex vivo (arrowhead in *Figure 6A$_8$*). As the dendrites reached the dorsolateral corner and explored locally, the change in shape appeared less pronounced (*Figure 6A$_9$*).

To quantitatively characterize the active targeting process, we categorized the bulk dendritic masses emanating from the main process according to their targeting directions: DL, DM, VM, and VL (*Figure 6B*). During the initial phase, the percentage of dendritic volume in each direction varied from 10% to 40% (*Figure 6C and D*), indicative of active exploration with little targeting specificity. Despite these variations, the total amount of dendritic mass seen in the VM direction over the entire imaging time (area under the graph of *Figure 6C*) was the smallest across all samples examined (*Figure 6E*). The initial phase of exploration in every direction was followed by a ~4 hr transitional phase during which DL1 PNs predominantly extended dendrites in 2 of the 4 directions (*Figure 6C*; *Figure 6—figure supplement 1D–E*). One of the 2 directions was always DL whereas the other was either DM or VL but never VM. In the final phase, DL1 PN dendrites always preferred DL out of the two available directions. Lastly, we analyzed the bulk dendritic movements. We defined bulk extension and retraction events when dendrites respectively extended and retracted more than 2 µm between two consecutive time frames. The analyses showed a striking shift from frequent extension and retraction towards stabilization, reflecting the pre- and post-targeting dynamics, respectively (*Figure 6F and G*).

Hence, long-term two-photon imaging of single-cell DL1 PNs revealed that dendrite targeting specificity increases over time via active targeting in a specific direction and stepwise elimination of unfavorable trajectory choices (see summary in *Figure 7F$_{1–3}$*).

## AO-LLSM imaging suggests a cellular mechanism underlying dendrite targeting specificity

To capture fast dynamics of single dendritic branches, we performed dual-color adaptive optical lattice sheet microscopy (AO-LLSM) imaging (*Chen et al., 2014*; *Wang et al., 2014*; *Liu et al., 2018*) of PNs every 30 s for 15 min, following a protocol we recently established (*Li et al., 2021*; *Li and Luo, 2021*). We selected 3 hr, 6 hr, and 12 hr APF pupal brains double-labeled with DL1 PNs and bulk adPNs (*Figure 7A–C*; *Figure 7—videos 1–3*). The labeling of adPNs with GFP outlined PN cell bodies and the developing antennal lobe but not the degenerating one, presumably because the GFP in larval-specific dendrites was quickly quenched upon glial phagocytosis (*Marin et al., 2005*).

In the 15 min imaging window, we observed four types of terminal branches regardless of neuronal types or developmental stages: (1) stable branch that existed throughout the entire imaging time,

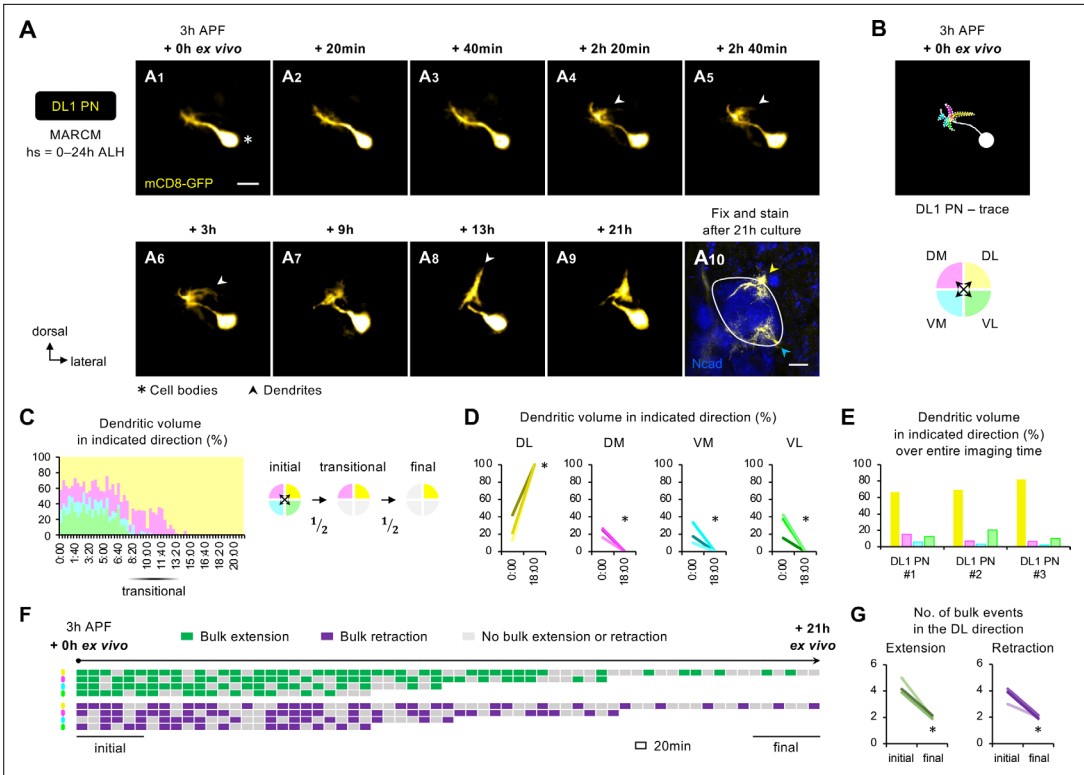

**Figure 6.** Two-photon time-lapse imaging reveals active dendrite targeting. (**A**) Two-photon time-lapse imaging of MARCM-labeled DL1 projection neuron (PN) (pseudo-colored in yellow) in a brain dissected at 3 hr APF and cultured for 21 hr ex vivo (**A**$_{1-9}$). Arrowheads in **A**$_{4-6}$ denote protrusions of dendritic branches towards the dorsolateral direction. After 21 hr culture, the explant was fixed and immuno-stained for N-Cadherin (Ncad; in blue) to outline the developing antennal lobe (**A**$_{10}$). Yellow and cyan arrowheads indicate DL1 PN dendrites and processes of other *GH146+* cells, respectively. (**B**) Neurite tracing of DL1 PN at the beginning of live imaging (3 hr APF + 0 hr ex vivo). Dendrites are categorized based on the directions to which they extend and color-coded accordingly. (**C**) Left: Quantification of the percentage of dendritic volume in indicated direction during the time-lapse imaging period reveals a transitional phase during which dendrites were found in only two out of the four directions. Right: Schematic of the initial, transitional, and final phases during the course of targeting. '½' denotes the reduction of available trajectory directions by half. Timestamp 00:00 refers to HH:mm; H, hour; m, minute. See *Figure 6—source data 1*. (**D**) Quantification of the percentage of DL1 PN dendritic volume in an indicated direction in 3 hr APF cultured brains at the beginning (0 hr ex vivo) and at/near the end of imaging (18 hr ex vivo). DL1 PN sample size = 3. *t*-test; *p<0.05. Timestamp 00:00 refers to HH:mm; H, hour; m, minute. (**E**) Quantification of the percentage of the sum of DL1 PN dendritic volume in indicated directions throughout the entire imaging time. DL1 PN sample size = 3. (**F**) Bulk dendrite dynamics of DL1 PN in *Figure 6A*. Each row represents bulk dendritic dynamics in the indicated direction (color-coded as in *Figure 6B*) across the 21 hr imaging period. Each block represents a 20 min window. Bulk extension (in green) and retraction (in magenta) events are defined as dendrites extending and retracting more than 2 μm between two consecutive time windows. The first and last six consecutive windows refer to the initial and final phases of imaging. (**G**) Quantification of the number of bulk extension and retraction events in the dorsolateral direction during the initial and final phases of imaging. DL1 PN sample size = 3. *t*-test; *p<0.05.

The online version of this article includes the following video, source data, and figure supplement(s) for figure 6:

**Source data 1.** Source data for *Figure 6C–G* and *Figure 6—figure supplement 1D and E*.

**Figure supplement 1.** Two-photon time-lapse imaging of DL1 projection neuron (PNs).

**Figure 6—video 1.** Two-photon time-lapse imaging of DL1 projection neuron (PN) dendrites.
https://elifesciences.org/articles/85521/figures#fig6video1

(2) transient branch that was produced and eliminated within the imaging window, (3) emerging branch that was produced after imaging began, and (4) retracting branch that was eliminated within the imaging period (*Figure 7—figure supplement 1A*). To examine if terminal branch dynamics exhibit any directional preference, we assigned the branches according to their targeting directions (*Figure 7D*). Extension and retraction events were defined when the speed exceeded 0.5 µm/min. Terminal branches were selected for analyses as branches closer to the main process were too dense to resolve. *Figure 7D$_{1-3}$* showed the dynamics of ~15 randomly selected terminal branches in each direction from the representative 3 hr, 6 hr, and 12 hr APF DL1 PNs (*Figure 7A–C*).

Quantitative analyses revealed that at 3 hr APF, DL1 PNs constantly produced, eliminated, extended, and retracted dendritic branches (*Figure 7A*, *Figure 7D$_1$*, *Figure 7—video 1*). Even stable branches were not immobile. Rather, they spent comparable amounts of time extending and retracting at ~1.5 µm/min (*Figure 7—figure supplement 1A$_1$, 1B*). Transient, emerging, and retracting branches had similar, but more variable speeds, ranging from 1 to 2.5 µm/min. Although there was no correlation between targeting direction and frequency/speed of extension/retraction, the number of stable branches in the VM direction was significantly lower than in other directions across all 3 hr DL1 PN samples examined (*Figure 7E$_1$*). This suggests that even though dendritic branches were developed in every direction at the early stages, those branches in the VM direction were short-lived and might be eliminated by retraction. The direction-dependent stability/lifespan of dendritic branches on the timescale of seconds uncovered from AO-LLSM imaging explains why bulk dendrites in unfavorable trajectories failed to persist in long-term two-photon imaging.

From 6 hr to 12 hr APF, DL1 PNs no longer manifested direction-specific branch de/stabilization (*Figure 7B–C*, *Figure 7D$_{2-3}$*, *Figure 7—videos 2–3*). At the same developmental stage, stable branches in one direction appeared indistinguishable from those in other directions in terms of abundance, frequency, and speed (*Figure 7D$_{2-3}$*, *Figure 7—figure supplement 1C–D*). This suggests that the entire dendritic mass tends to stay in equilibrium upon arrival at target regions. At 12 hr APF, the abundance of stable branches of DL1 PNs was the highest (*Figure 7D–E$_1$*). Also, the stable branches of 12 hr APF DL1 PNs moved at a significantly lower speed (~1 µm/min) (*Figure 7E$_2$*) and spent more time being stationary than those at 3 hr and 6 hr (*Figure 7—figure supplement 1B–D*). The reduced branch dynamics at 12 hr APF is consistent with observations from two-photon imaging showing fewer bulk extension/retraction events in the final phase of targeting (*Figure 6F–G*). Despite the slowdown, dendritic arborization was evident in terminal branches of 12 hr APF DL1 PNs (*Figure 7—figure supplement 1E*), suggesting that PN dendrites are transitioning from simple to complex branch architectures. Although it remains unclear if there is a causal relationship between reduced branch dynamics and increased structural complexity, we propose that both contribute to the sustentation of dendrite targeting specificity.

In summary, AO-LLSM imaging reveals that PNs selectively stabilize branches in the direction towards the target and destabilize those in the opposite direction, providing a cellular basis of dendrite targeting specificity. Upon arrival at the target, the specificity is sustained through branch stabilization in a direction-independent manner (summarized in *Figure 7F$_{4-7}$*).

## Embryonic-born PNs timely integrate into an adult olfactory circuit by simultaneous dendritic pruning and re-extension

In earlier sections, we uncovered wiring logic of larval-born PN dendritic patterning and cellular mechanisms of dendrite targeting specificity used to initiate olfactory map formation (*Figures 3–7*). In this final section, we focused on embryonic-born PNs, which participate in both larval and adult olfactory circuits by reorganizing their processes (*Marin et al., 2005*). Our previous study demonstrates that embryonic-born PNs prune their larval-specific dendrites during early metamorphosis (*Marin et al., 2005*; *Figure 1D$_{1-3}$*). Here, we examined when and how embryonic-born PNs re-extend dendrites used in the adult olfactory circuit.

It is known that γ neurons of *Drosophila* mushroom body (γ Kenyon cells) and sensory Class IV dendritic arborization (C4da) neurons prune their processes between 4 hr and 18 hr APF and show no signs of re-extension at 18 hr APF (*Lee et al., 2000*; *Watts et al., 2003*; *Lee et al., 2009*). Do embryonic-born adPNs follow a similar timeframe? We first examined developing brains double-labeled for embryonic-born DA4l/VA6/VA2 adPNs (collectively referred to as '*lov+* PNs') and early larval-born DC2 adPNs (*Figure 8A*; *Figure 8—figure supplement 1*). We found that, by 12 hr APF,

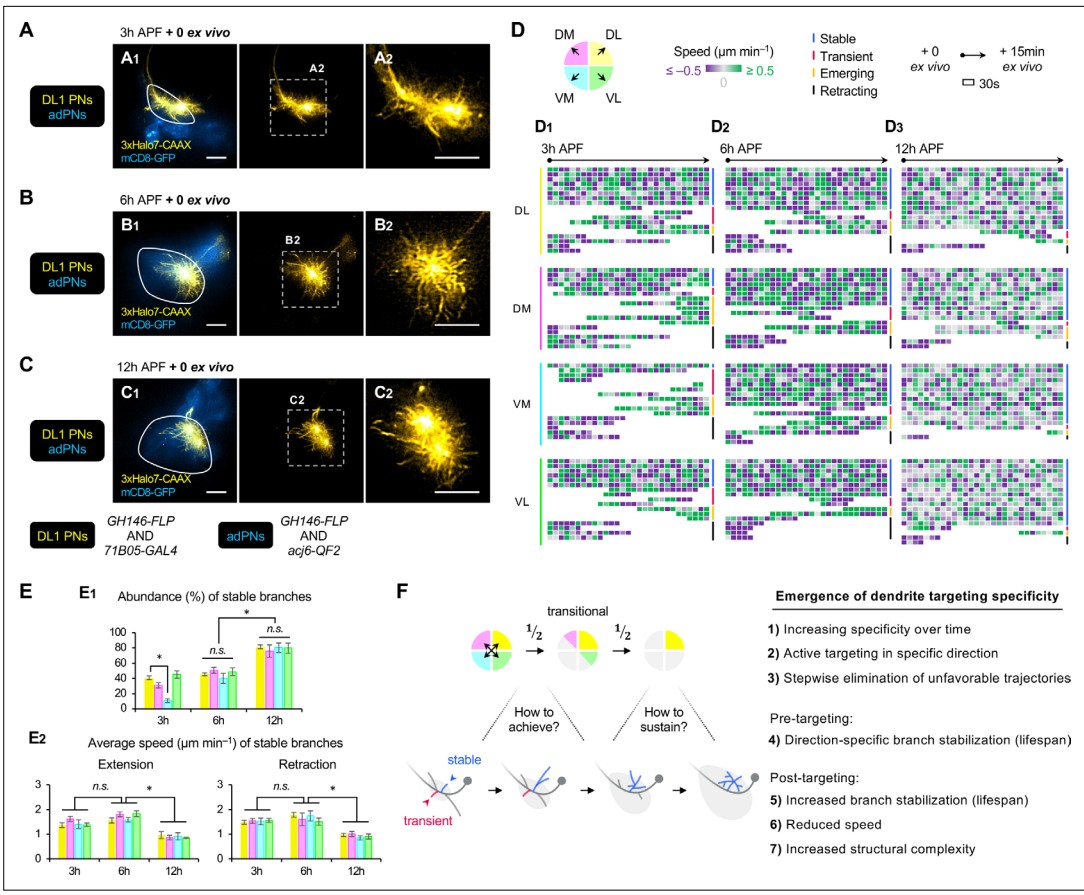

**Figure 7.** AO-LLSM time-lapse imaging reveals cellular mechanisms of dendrite targeting specificity. (**A–C**) AO-LLSM imaging of DL1 projection neurons (PNs) (*71B05+*; labeled in yellow) and anterodorsal projection neurons (adPNs) (*acj6+*; labeled in blue) in cultured brains dissected at 3 hr (**A**), 6 hr (**B**), and 12 hr (**C**) APF. Zoom-in, single z-section images of (**A₁**), (**B₁**), and (**C₁**) (outlined in dashed boxes) are shown in **A₂**, **B₂** and **C₂**, respectively. (**D**) Single dendritic branch dynamics of 3 hr (**D₁**), 6 hr (**D₂**), and 12 hr (**D₃**) DL1 PNs shown in **A–C**. Terminal branches are analyzed and categorized based on the directions in which they extend. Their speeds are color-coded using purple-gray-green gradients (negative speeds, retraction; positive speeds, extension). Individual branches are also assigned into four categories: stable, transient, emerging, and retracting (color-coded on the right; see *Figure 7—figure supplement 1A*). Each block represents a 30s window. Each row represents individual branch dynamics across the 15 min imaging period. (**E**) Quantification of the abundance (in percentage) of DL1 PN stable branches in indicated direction at 3 hr, 6 hr, and 12 hr (**E₁**). Average speed of DL1 PN stable branches in indicated direction at 3 hr, 6 hr, and 12 hr (**E₂**). DL1 PN sample size: 3 hr=4; 6 hr=3; 12 hr=3. Error bars, SEM; *t*-test; One-way ANOVA; *$p<0.05$; *n.s.*, $p≥0.05$. SEM, standard error of the mean; *n.s.*, not significant. See *Figure 7—source data 1*. (**F**) Summary of mechanisms underlying the emergence of dendrite targeting specificity revealed by two-photon and AO-LLSM imaging of DL1 PN dendrites.

The online version of this article includes the following video, source data, and figure supplement(s) for figure 7:

**Source data 1.** Source data for *Figure 7E*.

**Figure supplement 1.** Analyses of DL1 projection neuron (PN) dendritic branches captured by AO-LLSM imaging.

**Figure 7—video 1.** AO-LLSM time-lapse imaging of 3 hr DL1 projection neuron (PN) dendrites.
https://elifesciences.org/articles/85521/figures#fig7video1

**Figure 7—video 2.** AO-LLSM time-lapse imaging of 6 hr DL1 projection neuron (PN) dendrites.
https://elifesciences.org/articles/85521/figures#fig7video2

**Figure 7—video 3.** AO-LLSM time-lapse imaging of 12 hr DL1 projection neuron (PN) dendrites.
https://elifesciences.org/articles/85521/figures#fig7video3

*lov+* PNs already sent adult-specific dendrites to a region ventromedial to DC2 PN dendrites (green arrowhead in *Figure 8A₃*; see 3D rendering in *Figure 8—video 1*). This implies that *lov+* PNs have already caught up with DC2 PNs on dendrite development at this stage, and the re-extension of *lov+* PN dendrites must have happened even earlier. Indeed, we observed *lov+* PN dendrites innervated the developing antennal lobe extensively at 6 hr APF (*Figure 8A₂*). Such innervation was not observed at 0 hr APF (*Figure 8A₁*). After 12 hr APF, the time course of *lov+* PN dendrite development was comparable to that of DC2 PNs (*Figure 8A₄₋₆*).

To characterize dendritic re-extension at single-cell resolution, we developed a sparse, stochastic labeling strategy to label single *lov+* PNs (*Figure 8B*). We found that *lov+* PNs produced nascent branches from the main process dorsal to larval-specific dendrites as early as 3 hr APF (*Figure 8C₂₋₃*; arrowheads in *Figure 8C₆₋₇*). At 6 hr APF, when larval-specific dendrites were completely segregated from *lov+* PNs, the robust extension of adult-specific dendrites was seen across the developing antennal lobe (*Figure 8C₄*). These data indicate that *lov+* PNs re-extend their adult-specific dendrites at a more dorsal location before the larval-specific dendrites are completely pruned.

Do other embryonic-born PNs prune and re-extend their dendrites simultaneously? Like *lov* drivers, *Mz612-GAL4* labels embryonic-born PNs, one of which is VA6 PN (*Marin et al., 2005*). In 3 hr APF brains co-labeled for *Mz612+* and *lov+* PNs, we could unambiguously access three single embryonic-born PN types: (1) *lov+ Mz612–* PN, (2) *lov– Mz612+* PN, and (3) *lov+ Mz612+*PN (*Figure 8—figure supplement 2A–B*). Tracing of individual dendritic branches showed that all these PNs already re-extended dendrites to varying extents prior to the separation of larval-specific dendrites from the rest of the processes (*Figure 8—figure supplement 2C*). Thus, concurrent pruning and re-extension apply to multiple embryonic-born PN types.

To capture the remodeling at the higher temporal resolution, we performed two-photon time-lapse imaging of single embryonic-born PNs labeled by *Split7-GAL4* (*Figure 8D*, *Figure 8—video 2*, *Figure 8—figure supplement 3*). This *GAL4* labels one embryonic-born PN (either VA6 or VA2 PN) at early pupal stages but eight PN types at 24 hr APF (*Xie et al., 2021*). Initially (3 hr APF + 0 hr ex vivo), no adult-specific dendrites were detected in live *Split7+* PNs (*Figure 8D₁*). The following ~3 hr ex vivo saw thickening of the main process (arrowhead in *Figure 8D₃*). From 4 hr ex vivo onwards, re-extension occurred in the presumed developing antennal lobe located dorsal to larval-specific dendrites (arrowheads in *Figure 8D₄₋₈*; see traces in *Figure 8D₉*). Live imaging of *Split7+* PNs also revealed that fragmentation of larval-specific dendrites occurred at the distal ends (*Figure 8—figure supplement 3B₁₋₅*), and the process leading to larval-specific dendrites gradually disappeared as pruning approached completion (*Figure 8—figure supplement 3B₆₋₁₀*). These observations suggest that pruning of embryonic-born PN dendrites is not initiated by severing at the proximal end. Distal-to-proximal pruning, rather than in the reversed direction, further supports concurrent but spatially segregated pruning and re-extension processes.

It has been shown that dendritic pruning of embryonic-born PNs requires ecdysone signaling in a cell-autonomous manner (*Marin et al., 2005*). We asked if the re-extension process also depends on ecdysone signaling. We expressed a dominant negative form of ecdysone receptor (EcR-DN) in most PNs (including *lov+* PNs) and monitored the development of *lov+* PN dendrites (*Figure 8—figure supplement 4*). We found that inhibition of ecdysone signaling by *EcR-DN* expression not only suppressed pruning, but also blocked re-extension. This is consistent with a previous study reporting the dual requirement of ecdysone signaling in the pruning and re-extension of *Drosophila* anterior paired lateral (APL) neurons, although, unlike embryonic-born PNs, APL neurons prune and re-extend processes sequentially (at 6 hr and 18 hr APF, respectively) (*Mayseless et al., 2018*). We currently could not distinguish if the lack of re-extension is due to defective pruning, or if ecdysone signaling controls pruning and re-extension independently.

Taken together, our data demonstrate that embryonic-born PNs prune and re-extend dendrites simultaneously at spatially distinct regions, and that both processes require ecdysone signaling (*Figure 8E*). Such a 'multi-tasking' ability explains how embryonic-born PNs can re-integrate into the adult olfactory circuit and engage in its prototypic map formation in a timely manner.

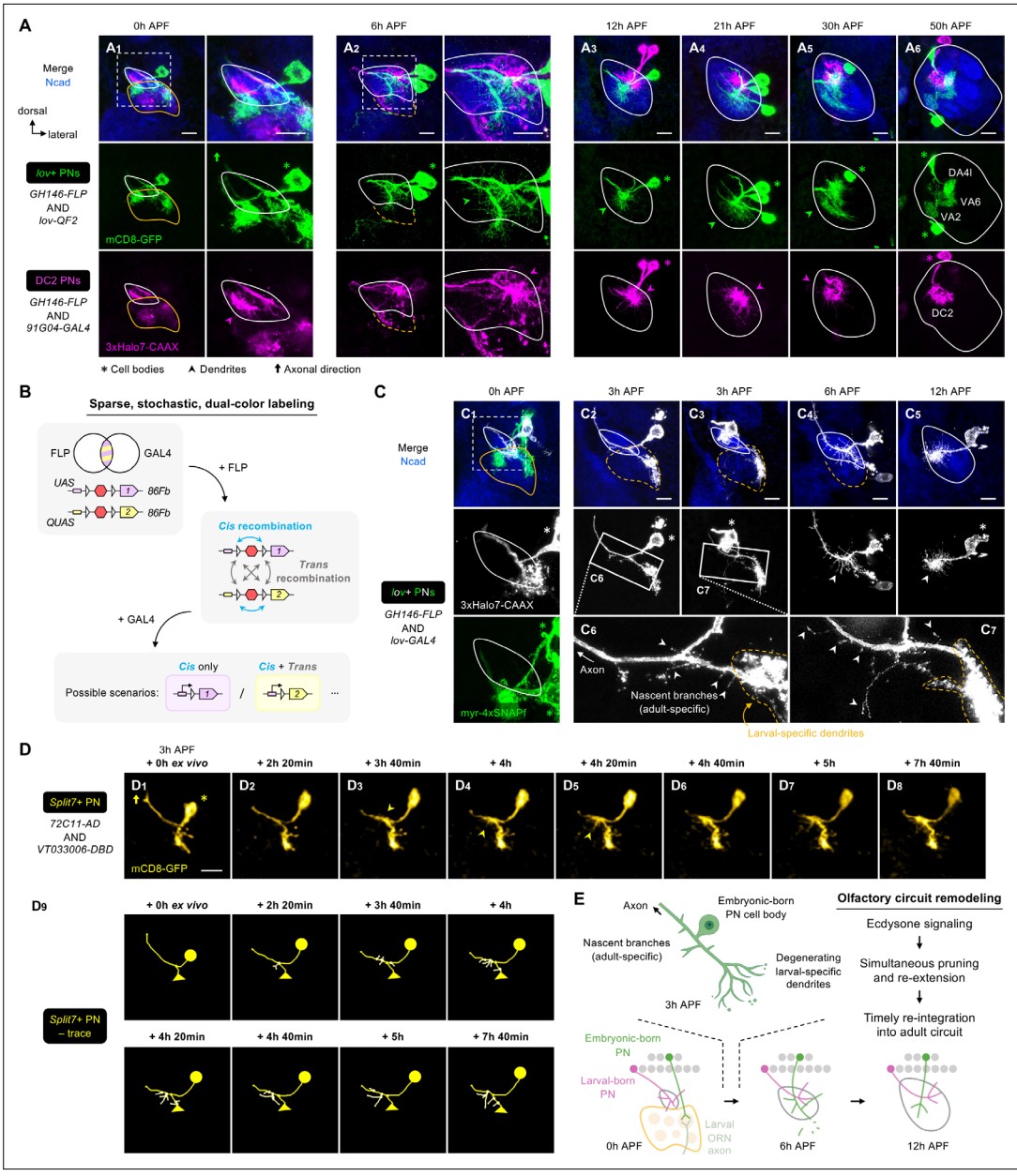

**Figure 8.** Embryonic-born projection neurons (PNs) timely participate in olfactory map formation via simultaneous pruning and re-extension. (**A**) Confocal images of fixed brains at indicated stages showing dendrite development of *lov+* PNs (embryonic-born; labeled in green) and *91G04+*DC2 PNs (larval-born; labeled in magenta). As *91G04-GAL4* also labels some embryonic-born PNs from 0 to 6 hr APF, their processes are found in the larval-specific antennal lobe (**A**₁, ₂). Right columns of **A**₁, ₂ show a zoom-in of the dashed boxes. Green arrowhead in (**A**₂) indicates robust dendrite re-extension of embryonic-born PNs across the developing antennal lobe at 6 hr APF. (**A**₁): N=6; (**A**₂): N=12; (**A**₃): N=9; (**A**₄): N=12; (**A**₅): N=9; (**A**₆): N=5. (**B**) Schematic of the sparse, stochastic, and dual-color labeling strategy. In this strategy, the same cell has one copy of *UAS*-responsive conditional reporter 1 and one copy of *QUAS*-responsive reporter 2, both of which are integrated into the same *86Fb* genomic locus (i.e. *UAS-FRT-stop-FRT-reporter1/QUAS-FRT-stop-FRT-reporter2*). FLP expression yields *cis* and *trans* recombination of *FRT* sites in a stochastic manner. Upon GAL4 expression, reporter 1 is expressed in cells with *cis* recombination, whereas reporter 2 is expressed only when *cis* and *trans* recombination events co-occur. (**C**) Sparse labeling of *lov+* PNs (labeled in green; single-cell *lov+* PNs in gray) at indicated developmental stages. (**C**₆) and (**C**₇) are zoom-in images of the rectangular boxes in (**C**₂) and (**C**₃), respectively. Arrowheads indicate nascent, adult-specific dendrites. Larval-specific dendrites are outlined by dashed orange lines. Arrows indicate axons projecting towards high brain centers. (**C**₁): N=6; (**C**₂–₃): N=6; (**C**₄): N=4; (**C**₅): N=4. (**D**) Two-photon time-lapse imaging of a single

*Figure 8 continued on next page*

*Figure 8 continued*

embryonic-born PN (*Split7+*; pseudo-colored in yellow) in a brain dissected at 3 hr APF and cultured for 23 hr ex vivo. Arrowhead in ($D_3$) denote the thickening of the main process. Arrowheads in $D_{4,5}$ denote dendritic protrusions dorsal to larval-specific dendrites. ($D_9$) shows neurite tracing of the embryonic-born PN. Triangles in ($D_9$) indicate the degenerating larval-specific dendrites. N=3. (**E**) Schematic summary of remodeling of embryonic-born PN dendrites. Following simultaneous pruning and re-extension, embryonic-born PNs timely integrate into an adult olfactory circuit and, together with larval-born PNs, participate in the prototypic map formation.

The online version of this article includes the following video and figure supplement(s) for figure 8:

**Figure supplement 1.** Dendrite development of *lov+* embryonic-born projection neurons (PNs).

**Figure supplement 2.** Dendrite re-extension of *lov+* and *Mz612+* embryonic-born projection neurons (PNs).

**Figure supplement 3.** Two-photon time-lapse imaging of *Split7+* projection neuron (PN) dendrites.

**Figure supplement 4.** Dual requirement of ecdysone signaling in pruning and re-extension of embryonic-born projection neuron (PN) dendrites.

**Figure 8—video 1.** 3D rendering of *z* stacks of indicated projection neurons (PNs) in 12 hr APF antennal lobe. https://elifesciences.org/articles/85521/figures#fig8video1

**Figure 8—video 2.** Two-photon time-lapse imaging of *Split7+* projection neuron (PN) dendrites. https://elifesciences.org/articles/85521/figures#fig8video2

## Discussion
## Wiring logic for the prototypic olfactory map

Prior to this study, no apparent logic linking PN lineage, birth order, and adult glomerular position has been found. Our systematic analyses of dendritic patterning at the resolution of specific PN types across development identified wiring logic underlying the spatial organization of the prototypic olfactory map (*Figures 3 and 4*).

We found that PNs of a given lineage and temporal cohort share similar dendrite targeting specificity and timing. Notably, dendrites of adPNs and lPNs respectively pattern the antennal lobe in rotating and binary manners following birth order. Based on our new observations and previous findings, we discuss possible mechanisms that execute the wiring logic to form the initial map: (1) specification of the initial dendrite targeting through combinatorial inputs from lineage and birth order, (2) PN dendrite-dendrite interactions, and (3) contribution of the degenerating larval-specific antennal lobe.

The spatial distinctions of cell bodies (e.g. *Figure 1D₁*), axons (e.g. *Figure 1—figure supplement 2A*), and dendrites (e.g. *Figure 4A₁*) of adPNs and lPNs observed in 0 hr APF pupal brain suggest that lineage endows projection specificity very early on. Lineage-specific transcription factors have been identified to instruct PN neurite targeting (*Komiyama et al., 2003*; *Komiyama and Luo, 2007*; *Li et al., 2017*; *Xie et al., 2022*), which might explain the differences between the adPN and lPN dendritic maps. Nonetheless, lineage alone does not account for the characteristic dendritic patterns. Rather, dendrite targeting can be predicted using combinatorial inputs from lineage and birth order. This combinatorial strategy is also seen in neuronal fate diversification and wiring of the *Drosophila* optic lobe and ventral nerve cord (*Erclik et al., 2017*; *Mark et al., 2021*), suggesting that it is a general principle in wiring the fly brain and likely also used in vertebrates (*Holguera and Desplan, 2018*; *Sen, 2023*). Substantial advances have been made in understanding how temporal patterning arises for intra-lineage specification (*Doe, 2017*; *Miyares and Lee, 2019*). For instance, the embryonic ventral nerve cord neuroblasts sequentially express a cascade of temporal transcription factors (TTFs) to specify temporal identity (*Isshiki et al., 2001*). Larval optic lobe neuroblasts also deploy the same strategy but use a completely different TTF cascade (*Li et al., 2013*). Earlier studies show Chinmo, a TTF, and RNA-binding proteins that regulate Chinmo translation, control neuronal cell fate of the adPN lineage (*Zhu et al., 2006*; *Liu et al., 2015*). Specifically, DL1 PNs mutant for Chinmo project dendrites to D glomerulus that is targeted by the fourth larval-born adPNs (*Zhu et al., 2006*), demonstrating temporal order specifies final glomerular targeting. However, whether approximate temporal cohorts of a given PN lineage we described arise from sequential expression of temporal factors, and how such factors translate into initial dendrite patterning remains a fertile ground for future studies.

Our time-lapse imaging data reveals robust PN dendritic dynamics during the initial targeting process (*Figures 5–8*), suggesting that cellular interactions among PN dendrites contribute to the initial map formation. This appears to contrast with the PN-ORN map in the mature antennal lobe, which is highly stable; connection specificity remains largely unchanged upon genetic ablation of their synaptic partners (*Berdnik et al., 2006*). Future works using early-onset genetic drivers for specific PN types for ablation can be used to investigate interactions between different PN groups, such as adPNs and lPNs, in the construction of the initial PN dendrite map.

Does the degenerating larval-specific antennal lobe contribute to the initial dendrite patterning of the developing adult-specific antennal lobe? Earlier studies found that the larval-specific ORN axons secrete semaphorins, Sema-2a and Sema-2b, which act as repulsive ligands for dendrites of Sema-1a-expressing PNs (including DL1 PNs) (*Komiyama et al., 2007*; *Sweeney et al., 2011*). As the larval-specific lobe is located ventromedial to the adult-specific lobe, Sema-2a/b and Sema-1a form opposing gradients along the dorsolateral-ventromedial axis. When DL1 PNs (the first-born/ developed) begin to target their dendrites, this repulsive action could destabilize branches in the ventromedial direction and thus favor dorsolateral targeting. This provides a plausible explanation as to why the adPN rotation pattern begins at the dorsolateral position. It would be interesting to see if the pattern is perturbed upon ablation of larval-specific ORNs.

Our new tools for labeling and genetic manipulation of distinct PN types (*Figure 2*) will now enable in-depth investigations into the potential cellular interactions and molecular mechanisms leading to the initial map organization.

## Wiring logic evolves as development proceeds

After the initial map formation at 12 hr APF, dendrite positions in the antennal lobe could change substantially in the next 36 hr (for example, see DC2 PNs in *Figure 3B$_{4-6}$* and DA1 and VA1d/DC3 PNs in *Figure 3C$_{4-7}$*). These changes occur when dendrites of PNs with neighboring birth order begin to segregate and when ORN axons begin to invade the antennal lobe. Accordingly, the ovoid-shaped antennal lobe turns into a globular shape (30–50 hr APF; *Figure 3C$_{6-7}$*). These PN-autonomous and non-autonomous changes likely mask the initial wiring logic, explaining why previous studies, which mostly focused on examining the final glomerular targets in adults (*Jefferis et al., 2001*), have missed the earlier organization. Interestingly, the process of PN dendritic segregation coincides with the peak of PN transcriptomic diversity at 24 hr APF (*Li et al., 2017*; *Xie et al., 2021*).

Recent proteomics and genetic analyses have indicated that PN dendrite targeting is mediated by cell-surface proteins cooperating as a combinatorial code (*Xie et al., 2022*). The evolving wiring logic, which is consistent with the stepwise assembly of an olfactory circuit (*Hong and Luo, 2014*), suggests the combinatorial codes are not static. We propose that PNs use a numerically simpler code for initial dendrite targeting. Following the expansion of transcriptomic diversity, PNs acquire a more complex code mediating dendritic segregation of neighboring PNs and matching of PN dendrites and ORN axons. Functional characterization of differentially expressed genes between 12 hr and 24 hr APF PNs may provide molecular insights into how the degree of discreteness in the olfactory map arises.

Although the initial wiring logic is not apparent in the final map, several lines of evidence suggest the final map depends on the initial map. First, as mentioned above, the change of the temporal identity of DL1 PNs affects glomerular targeting (*Zhu et al., 2006*). Second, loss of Sema-1a in DL1 PNs occasionally causes mistargeting in areas outside of the antennal lobe, and dendrite mistargeting phenotype along the dorsolateral-ventromedial axis is persistent across development as well as in adulthood (*Komiyama et al., 2007*). Our work thus demonstrates that identification of the wiring logic in the early stages should help us better resolve the architectures in complex neural circuits.

## Selective branch stabilization as a cellular mechanism for dendrite targeting

Utilizing an early pupal brain explant culture system coupled with two-photon and AO-LLSM imaging (*Figure 5*), we presented the first time-lapse videos following dendrite development of a specific PN type – DL1 PNs (*Figures 6 and 7*). We found that DL1 PN dendrites initiate active targeting towards their dorsolateral target with direction-dependent branch stabilization. This directional selectivity provides a cellular basis for the emerging targeting specificity of PN dendrites at the beginning of olfactory map formation.

Although selective branch stabilization as a mechanism to achieve axon targeting specificity has been described in neurons in the vertebrate and invertebrate systems (e.g. *Yates et al., 2001*; *Li et al., 2021*), our time-lapse imaging showed, for the first time to our knowledge, that selective branch stabilization is also used to achieve dendrite targeting specificity. Furthermore, AO-LLSM imaging revealed that selective stabilization and destabilization of dendritic branches occur on the timescale of seconds. As the rate of olfactory circuit development in the brain explants was slower than normal development (*Figure 5F*), we might have captured PN dendritic dynamics in slow motion. Using AO-LLSM for high spatiotemporal resolution imaging, we just begin to appreciate how fast PN dendrites are coordinating trajectory choices with branch stabilization to make the appropriate decision. Having characterized the dendritic branch dynamics of the wild-type DL1 PNs, we have set the stage for future studies addressing how positional cues and the downstream signaling instruct wiring, and whether other PN types follow similar rules as DL1 PNs.

## Simultaneous pruning and re-extension as novel remodeling mechanism for neuronal remodeling

Our data on embryonic-born adPN dendrite development reveals a novel mode of neuronal remodeling during metamorphosis (*Figure 8*). In mushroom body γ neurons and body wall somatosensory neurons, two well-characterized systems, larval-specific neurites are first pruned, followed by re-extension of adult-specific processes (*Watts et al., 2003*; *Williams and Truman, 2005*; *Yaniv and Schuldiner, 2016*). However, embryonic-born adPNs prune larval-specific dendrites and re-extend adult-specific dendrites simultaneously but at spatially separated subcellular compartments. Such spatial segregation suggests that regional external cues could elicit compartmentalized downstream signals leading to opposite effects on the dendrites. Subcellular compartmentalization of signaling and cytoskeletal organization has been observed in diverse neuron types across species (*Rolls et al., 2007*; *Kanamori et al., 2013*; *O'Hare et al., 2022*).

Why do embryonic-born adPNs 'rush' to re-extend dendrites? During normal development, it takes at least 18 hr for embryonic-born adPNs to produce and properly target dendrites (growth at 3–6 hr APF, initial targeting at 6–12 hr APF, and segregation at 21–30 hr APF). Given that the dendritic re-extension of embryonic-born PNs is ecdysone dependent (*Figure 8—figure supplement 4*), if the PNs did not re-extend dendrites at 3 hr APF, they would have to wait for the next ecdysone surge at ~20 hr APF (*Thummel, 2001*), which might be too late for their dendrites to engage in the prototypic map formation. Thus, embryonic-born PNs develop a remodeling strategy that coordinates with the timing of systemic ecdysone release. By simultaneous pruning and re-extension, embryonic-born adPNs timely re-integrate into the adult prototypic map that readily serves as a target for subsequent ORN axon innervation.

In conclusion, our study highlights the power and necessity of type-specific neuronal access and time-lapse imaging to identify wiring logic and mechanisms underlying the origin of an olfactory map. Applying similar approaches to other developing neural maps across species should broaden our understanding of the generic and specialized designs that give rise to functional maps with diverse architectures.

## Materials and methods
### *Drosophila* stocks and husbandry

Flies were maintained on a standard cornmeal medium at 25 °C. Fly lines used in this study included *GH146-FLP* (*Hong et al., 2009*), *QUAS-FRT-stop-FRT-mCD8-GFP* (*Potter et al., 2010*), *UAS-mCD8-GFP* (*Lee and Luo, 1999*), *UAS-mCD8-FRT-GFP-FRT-RFP* (*Stork et al., 2014*), *VT033006-GAL4* (*Tirian and Dickson, 2017*), *Mz19-GAL4* (*Jefferis et al., 2004*), *91G04-GAL4* (*Jenett et al., 2012*), *Mz612-GAL4* (*Marin et al., 2005*), *71B05-GAL4* (*Jenett et al., 2012*), *Split7-GAL4* (*Xie et al., 2021*), *QUAS-FLP* (*Potter et al., 2010*), and *UAS-EcR.B1-ΔC655.F645A* (*Cherbas et al., 2003*). The following *GAL4* lines were obtained from Bloomington *Drosophila* Stock Center (BDSC): *tsh-GAL4* (BDSC #3040) and *lov-GAL4* (BDSC #3737).

The following two stocks were used for MARCM analyses: (1) *UAS-mCD8-GFP, hs-FLP; FRT$^{G13}$, tub-GAL80;; GH146-GAL4*, and (2) *FRT$^{G13}$, UAS-mCD8-GFP* (*Lee and Luo, 1999*).

The following lines were generated in this study: *UAS-FRT^10-stop-FRT^10-3xHalo7-CAAX* (on either II or III chromosome), *UAS-FRT-myr-4xSNAPf-FRT-3xHalo7-CAAX* (III), *UAS-FRT-myr-mGreenLantern-FRT-3xHalo7-CAAX* (II), *QUAS-FRT-stop-FRT-myr-4xSNAPf* (III), *run-T2A-FLP* (X), *acj6-T2A-FLP* (X), *acj6-T2A-QF2* (X), *CG14322-T2A-QF2* (III), and *lov-T2A-QF2* (II).

### *Drosophila* genotypes

*Figure 1D*, *Figure 1—figure supplement 1*, *Figure 1—figure supplement 2*: *run-T2A-FLP/+; UAS-mCD8-FRT-GFP-FRT-RFP/+; VT033006-GAL4/+*

*Figure 3A*: *acj6-T2A-QF2/+; GH146-FLP, QUAS-FRT-stop-FRT-mCD8-GFP/UAS-FRT^10-stop-FRT^10-3xHalo7-CAAX; 71B05-GAL4/+*

*Figure 3B*, *Figure 3—figure supplement 1C*: *GH146-FLP/UAS-FRT^10-stop-FRT^10-3xHalo7-CAAX; 91G04-GAL4/CG14322-T2A-QF2, QUAS-FRT-stop-FRT-myr-4xSNAPf*

*Figure 3C*: *acj6-T2A-FLP/+; Mz19-GAL4; UAS-FRT-myr-4xSNAPf-FRT-3xHalo7-CAAX/+*

*Figure 3D*, *Figure 3—figure supplement 2*, *Figure 3—figure supplement 3*: *UAS-mCD8-GFP, hs-FLP/+; FRT^G13, tub-GAL80/FRT^G13, UAS-mCD8-GFP;; GH146-GAL4 (IV)/+*

*Figure 3—figure supplement 1A*: *GH146-FLP/UAS-FRT^10-stop-FRT^10-3xHalo7-CAAX; 71B05-GAL4/+*

*Figure 3—figure supplement 1B*: *GH146-FLP/UAS-FRT^10-stop-FRT^10-3xHalo7-CAAX; 91G04-GAL4/+*

*Figure 3—video 1*: Please refer to *Figure 3* for genotypes.

*Figure 4A*, *Figure 4—figure supplement 1*: *GH146-FLP, UAS-FRT^10-stop-FRT^10-3xHalo7-CAAX/tsh-GAL4; CG14322-T2A-QF2, QUAS-FRT-stop-FRT-myr-4xSNAPf/+*

*Figure 4B*: *UAS-mCD8-GFP, hs-FLP/+; FRT^G13, tub-GAL80/FRT^G13, UAS-mCD8-GFP;; GH146-GAL4 (IV)/+*

*Figure 8—figure supplement 2*: *acj6-T2A-FLP/+; tsh-GAL4, UAS-mCD8-FRT-GFP-FRT-RFP*

*Figure 4—video 1*: Please refer to *Figure 4* for genotypes.

*Figure 5E*, *Figure 5—video 1*: *run-T2A-FLP/+; UAS-FRT-myr-mGreenLantern-FRT-3xHalo7-CAAX/+; VT033006-GAL4/+*

*Figure 5F*: *UAS-mCD8-GFP/+; VT033006-GAL4/+*

*Figure 5G1*: *GH146-FLP/UAS-FRT^10-stop-FRT^10-3xHalo7-CAAX; 71B05-GAL4/+*

*Figure 5G2*: *GH146-FLP/tsh-GAL4; UAS-FRT^10-stop-FRT^10-3xHalo7-CAAX/+*

*Figure 5—figure supplement 1*, *Figure 5—video 2*: *acj6-T2A-FLP/+; Mz19-GAL4/ UAS-FRT-myr-mGreenLantern-FRT-3xHalo7-CAAX*

*Figure 6A*, *Figure 6—figure supplement 1*, *Figure 6—video 1*: *UAS-mCD8-GFP, hs-FLP/+; FRT^G13, tub-GAL80/FRT^G13, UAS-mCD8-GFP;; GH146-GAL4 (IV)/+*

*Figure 7A–C*, *Figure 7—figure supplement 1*, *Figure 7—videos 1–3*: *acj6-T2A-QF2/+; GH146-FLP, QUAS-FRT-stop-FRT-mCD8-GFP/UAS-FRT^10-stop-FRT^10-3xHalo7-CAAX; 71B05-GAL4/+*

*Figure 8A*, *Figure 8—figure supplement 1*: *GH146-FLP, QUAS-FRT-stop-FRT-mCD8-GFP/lov-T2A-QF2; UAS-FRT^10-stop-FRT^10-3xHalo7-CAAX/91G04-GAL4*

*Figure 8C*: *GH146-FLP/lov-GAL4; UAS-FRT^10-stop-FRT^10-3xHalo7-CAAX/ QUAS-FRT-stop-FRT-myr-4xSNAPf*

*Figure 8D*, *Figure 8—figure supplement 3*, *Figure 8—video 2*: *UAS-mCD8-GFP/+; Split7-GAL4 (i.e. FlyLight SS01867: 72C11-p65ADZp; VT033006-ZpGDBD)/+*

*Figure 8—figure supplement 2*: *GH146-FLP, QUAS-FRT-stop-FRT-mCD8-GFP/lov-T2A-QF2, Mz612-GAL4; UAS-FRT^10-stop-FRT^10-3xHalo7-CAAX/+*

*Figure 8—figure supplement 4A*: *lov-T2A-QF2, QUAS-FLP/+; VT033006-GAL4/ UAS-mCD8-FRT-GFP-FRT-RFP*

*Figure 8—figure supplement 4B*: *lov-T2A-QF2, QUAS-FLP/UAS-EcR-DN; VT033006-GAL4/ UAS-mCD8-FRT-GFP-FRT-RFP*

*Figure 8—video 1*: Please refer to *Figure 8* for genotypes.

## MARCM clonal analyses

MARCM clonal analyses have been previously described (*Lee and Luo, 1999*). Larvae of the genotype *UAS-mCD8-GFP, hs-FLP/+; FRT$^{G13}$, tub-GAL80/FRT$^{G13}$, UAS-mCD8-GFP;; GH146-GAL4/+* were heat shocked at 37 °C for 1 hr. To label the first-born DL1 PNs, heat shock was applied at 0–24 hr after larval hatching (ALH). MARCM clones of early, middle (mid-late for adPNs), and late larval-born PNs were generated by applying heat shocks at 42–48 hr, 66–72 hr, and 96–100 hr ALH, respectively. As larvae developed at different rates (*Tennessen and Thummel, 2011*), we reasoned that even if we could collect 0 hr–2 hr ALH larvae, their development might have varied by the time of heat shock. To minimize the effects of unsynchronized development, we selected those heat-shocked larvae that were among the first to form puparia and collected these white pupae in a ~3 hr window for the clonal analyses.

## Transcriptomic analyses

Transcriptomic analyses have been described previously (*Xie et al., 2021*). tSNE plots and dot plots were generated in Python using PN single-cell RNA sequencing data and code available at https://github.com/Qijing-Xie/FlyPN_development (*Xie, 2021*).

## Generation of *T2A-QF2/FLP* lines

To generate a *T2A-QF2/FLP* donor vector for *acj6* (we used the same strategy for *run, CG14322* and *lov*), a ~2000 bp genomic sequence flanking the stop codon of *acj6* was PCR amplified and introduced into *pCR-Blunt II-TOPO* (ThermoFisher Scientific #450245), forming *pTOPO-acj6*. To build *pTopo-acj6-T2A-QF2*, *T2A-QF2* including *loxP*-flanked *3xP3-RFP* was PCR amplified from *pBPGUw-HACK-QF2* (Addgene #80276), followed by insertion into *pTOPO-acj6* right before the stop codon of *acj6* by DNA assembly (New England BioLabs #E2621S). To generate *T2A-FLP*, we PCR-amplified *FLP* from the genomic DNA of *GH146-FLP* strain. *QF2* in *pTopo-acj6-T2A-QF2* was then replaced by *FLP* through DNA assembly. Using CRISPR Optimal Target Finder (*Gratz et al., 2014*), we selected a 20 bp gRNA target sequence that flanked the stop codon and cloned it into *pU6-BbsI-chiRNA* (Addgene #45946). If the gRNA sequence did not flank the stop codon, silent mutations were introduced at the PAM site of the donor vector by site-directed mutagenesis. Donor and gRNA vectors were co-injected into *Cas9* embryos in-house or through BestGene.

## Generation of FLP-out reporters

To generate *pUAS-FRT$^{10}$-stop-FRT$^{10}$-3xHalo7-CAAX*, *FRT$^{10}$-stop-FRT$^{10}$* was PCR amplified from *pUAS-FRT$^{10}$-stop-FRT$^{10}$-mCD8-GFP* (*Li et al., 2021*) and inserted into *pUAS-3xHalo7-CAAX* (Addgene #87646) through NotI and DNA assembly.

To generate *pUAS-FRT-myr-4xSNAPf-FRT-3xHalo7-CAAX*, we first PCR amplified *myr-4xSNAPf* from *pUAS-myr-4xSNAPf* (Addgene #87637) using *FRT*-containing primers. *FRT-myr-4xSNAPf-FRT* was then introduced into *pCR-Blunt II-TOPO*, forming *pTOPO-FRT-myr-4xSNAPf-FRT*. Using NotI-containing primers, *FRT-myr-4xSNAPf-FRT* was PCR amplified and subcloned into *pUAS-3xHalo7-CAAX* through NotI.

To generate *pUAS-FRT-myr-mGreenLantern-FRT-3xHalo7-CAAX*, we first PCR amplified *mGreen-Lantern* from *pcDNA3.1-mGreenLantern* (Addgene #161912). Using MluI and XbaI, we replaced *4xSNAPf* in *pUAS-myr-4xSNAPf* with *mGreenLantern* to build *pUAS-myr-mGreenLantern*. *myr-mGreenLantern* was PCR amplified with the introduction of *FRT* sequence, followed by insertion into *pCR-Blunt II-TOPO*. Using the NotI-containing primers, *FRT-myr-mGreenLantern-FRT* was PCR ampli-fied and subcloned into *pUAS-3xHalo7-CAAX* through NotI.

To generate *pQUAS-FRT-stop-FRT-myr-4xSNAPf*, we first PCR amplified *FRT-stop* from *pJFRC7-20XUAS-FRT-stop-FRT-mCD8-GFP* (*Li et al., 2021*) and inserted it into *pTOPO-FRT-myr-4xSNAPf-FRT* through DNA assembly to form *pTOPO-FRT-stop-FRT-myr-4xSNAPf-FRT*. Using NotI-containing forward and KpnI-containing reverse primers, *FRT-stop-FRT-myr-4xSNAPf* was PCR amplified and subcloned into *p10XQUAST*. *p10XQUAST* was generated using *p5XQUAS* (Addgene #24349) and *p10xQUAS-CsChrimson* (Addgene #163629).

*attP24* and *86Fb* landing sites were used for site-directed integration.

## Immunofluorescence staining and confocal imaging

Fly brain dissection for immunostaining and live imaging has been described (*Wu and Luo, 2006*). Briefly, brains were dissected in phosphate-buffered saline (PBS) and fixed with 4%

paraformaldehyde in PBS for 20 min on a nutator at room temperature. Fixed brains were washed with 0.1% Triton X-100 in PBS (PBST) for 10 min twice. After blocking with 5% normal donkey serum in PBST for 1 hr at room temperature, the brains were incubated with primary antibodies overnight at 4 °C. After PBST wash, brains were incubated with secondary antibodies (1:1000; Jackson ImmunoResearch) in dark for 2 hr at room temperature. Washed and mounted brains were imaged with confocal laser scanning microscopy (ZEISS LSM 780; LSM 900 with Airyscan 2). Images were processed with ImageJ. Neurite tracing images were generated using Simple Neurite Tracer (SNT) (*Arshadi et al., 2021*). Primary antibodies used included chicken anti-GFP (1:1000; Aves Lab #GFP-1020), rabbit anti-DsRed (1:500; TaKaRa #632496), rat anti-Cadherin DN (1:30; Developmental Studies Hybridoma Bank DSHB DN-Ex#8 supernatant), and mouse anti-Bruchpilot (1:30; DSHB nc82 supernatant).

## Chemical labeling

Chemical labeling of *Drosophila* brains has been described (*Kohl et al., 2014*). Janelia Fluor (JF) Halo and SNAP ligands (stocks at 1 mM) were gifts from Dr. Luke Lavis (*Grimm et al., 2017*; *Grimm et al., 2021*).

Fixed brains were washed with PBST for 5 min, followed by incubation with Halo and/or SNAP ligands (diluted in PBS) for 45 min at room temperature. Brains were then washed with PBST for 5 min, followed by blocking and immunostaining if necessary. For the co-incubation of Halo and SNAP ligands, JF503-cpSNAP (1:1000) and JF646-Halo (1:1000) were used. Alternatively, JFX650-SNAP (1:1000) and JFX554-Halo (1:10,000) were used. When only Halo ligands were needed, either JF646-Halo or JF635-Halo (1:1000) was used.

For live brain imaging, dissected brains were incubated with Halo ligands diluted in culture media (described below) for 30 min at room temperature. For two-photon imaging, JF570-Halo was used at 1:5000. For AO-LLSM imaging, following JF646-Halo incubation at 1:1000, the brains were incubated with 1 μM Sulforhodamine 101 (Sigma) for 5 min at room temperature. The brains were then briefly washed with culture media before imaging.

## Brain explant culture setup and medium preparation

Brain explant culture setup was modified based on *Li et al., 2021*; *Li and Luo, 2021*. A Sylgard plate with a thickness of ~2 millimeters was prepared by mixing base and curing agent at 10:1 ratio (DOW SYLGARD 184 Silicone Elastomer Kit). The mixture was poured into a 60 mm × 15 mm dish in which it was cured for two days at room temperature. Once cured, the plate was cut into small squares (~15 mm × ~15 mm). Indentations were created based on the size of an early pupal brain using a No.11 scalpel. Additional slits were made around the indentations for attaching imaginal discs which served as anchors to hold the brain position. A square Sylgard piece was then placed in a 60 mm × 15 mm dish or on a 25 mm round coverslip in preparation for two-photon/AO-LLSM imaging.

Culture medium was prepared based on published methods (*Rabinovich et al., 2015*; *Li and Luo, 2021*; *Li et al., 2021*). The medium contained Schneider's *Drosophila* Medium (ThermoFisher Scientific #21720001), 10% heat-inactivated Fetal Bovine Serum (ThermoFisher Scientific #16140071), 10 μg/mL human recombinant insulin (ThermoFisher Scientific #12585014; stock = 4 mg/mL), 1:100 Penicillin-Streptomycin (ThermoFisher Scientific #15140122). For 0 hr–6 hr APF brain culture, 0.5 mM ascorbic acid (Sigma #A4544; stock concentration = 50 mg/mL in water) was included. 20-hydroxyecdysone (Sigma #H5142; stock concentration = 1 mg/mL in ethanol) was used for 0 hr–6 hr and 12 hr brain explants at 20 μM and 2 μM, respectively. Culture medium was oxygenated for 20 min before use.

## Single- and dual-color imaging with two-photon microscopy

Single- and dual-color imaging of PNs were performed at room temperature using a custom-built two-photon microscope (Prairie Technologies) with a Chameleon Ti:Sapphire laser (Coherent) and a 16 X water-immersion objective (0.8 NA; Nikon). Excitation wavelength was set at 920 nm for GFP imaging, and at 935 nm for co-imaging of mGreenLantern and JF570-Halo. z-stacks were obtained at 4 μm increments (10 μm increments for *Figure 5—video 1*). Images were acquired at a resolution of 1024 × 1024 pixel$^2$ (512 × 512 for *Figure 5—video 1*), with a pixel dwell time of 6.8 μs and an optical zoom of 2.1, and at a frequency every 20 min for 8–23 hr.

## Dual-color imaging with AO-LLSM

For AO-LLSM-based imaging, the excitation and detection objectives along with the 25 mm coverslip were immersed in ~40 mL of culture medium at room temperature. Explant brains held on Sylgard plate were excited simultaneously using 488 nm (for GFP) and 642 nm (for JF-646) lasers operating with ~2–10 mW input power to the microscope (corresponding to ~10–50 µW at the back aperture of the excitation objective). An exposure time of 20–50 msec was used to balance imaging speed and signal-to-noise ratio (SNR). Dithered lattice light-sheet patterns with an inner/outer numerical aperture of 0.35/0.4 or 0.38/0.4 were used. The optical sections were collected by an axial step size of 250 nm in the detection objective coordinate, with a total of 81–201 steps (corresponding to a total axial scan range of 20–50 µm). Emission light from GFP and JF-646 was separated by a dichromatic mirror (Di03-R561, Semrock, IDEX Health & Science, LLC, Rochester, NY) and captured by two Hamamatsu ORCA-Fusion sCMOS cameras simultaneously (Hamamatsu Photonics, Hamamatsu City, Japan). Prior to the acquisition of the time series data, the imaged volume was corrected for optical aberrations using a two-photon guide star-based adaptive optics method (*Chen et al., 2014*; *Wang et al., 2014*; *Liu et al., 2018*). Each imaged volume was deconvolved using Richardson-Lucy algorithm on HHMI Janelia Research Campus' or Advanced Bioimaging Center's computing cluster (https://github.com/scopetools/cudadecon, *Lambert et al., 2023*; https://github.com/abcucberkeley/LLSM3DTools, *Ruan and Upadhyayula, 2020*) with experimentally measured point spread functions obtained from 100 or 200 nm fluorescent beads (Invitrogen FluoSpheres Carboxylate-Modified Microspheres, 505/515 nm, F8803, FF8811). The AO-LLSM was operated using a custom LabVIEW software (National Instruments, Woburn, MA).

## Statistics

For data analyses, *t*-test and one-way ANOVA were used to determine *p* values as indicated in the figure legend for each graph, and graphs were generated using Excel. Exact *p* values were provided in source data files.

## Material and data availability

All reagents generated in this study are available from the lead corresponding author upon request. *Figure 3—figure supplement 3—source data 1*, *Figure 6—source data 1*, and *Figure 7—source data 1* contain the numerical and statistical data used to generate the figures. The confocal imaging dataset is available at Brain Image Library under DOI https://doi.org/10.35077/g.933.

## Acknowledgements

We thank the Luo lab members for constructive feedback on the manuscript; Tzumin Lee for sharing equipment at Janelia Research Campus; Luke Lavis for sharing JF dyes. This work was supported by a grant from NIH (R01 DC005982 to LL). TL was supported by NIH 1K99DC01883001. GL and SU are funded by Philomathia Foundation. SU is funded by the Chan Zuckerberg Initiative Imaging Scientist program. SU is a Chan Zuckerberg Biohub Investigator. EB and LL are HHMI investigators.

## Additional information

### Funding

| Funder | Grant reference number | Author |
| --- | --- | --- |
| National Institutes of Health | R01 DC005982 | Liqun Luo |
| Philomathia Foundation | | Gaoxiang Liu<br>Srigokul Upadhyayula |
| Chan Zuckerberg Initiative | | Srigokul Upadhyayula |
| National Institutes of Health | 1K99DC01883001 | Tongchao Li |

| Funder | Grant reference number | Author |
|---|---|---|
| Howard Hughes Medical Institute | | Eric Betzig Liqun Luo |

The funders had no role in study design, data collection and interpretation, or the decision to submit the work for publication.

## Author contributions

Kenneth Kin Lam Wong, Conceptualization, Data curation, Formal analysis, Investigation, Visualization, Methodology, Writing - original draft; Tongchao Li, Resources, Investigation, Methodology, Writing – review and editing; Tian-Ming Fu, Gaoxiang Liu, Resources, Data curation, Investigation, Methodology, Writing – review and editing; Cheng Lyu, Resources, Methodology, Writing – review and editing; Sayeh Kohani, Data curation, Investigation; Qijing Xie, Data curation, Investigation, Writing – review and editing; David J Luginbuhl, Resources, Data curation, Writing – review and editing; Srigokul Upadhyayula, Eric Betzig, Resources, Supervision, Writing – review and editing; Liqun Luo, Conceptualization, Supervision, Funding acquisition, Investigation, Methodology, Project administration, Writing – review and editing

## Author ORCIDs

Kenneth Kin Lam Wong  http://orcid.org/0000-0002-5597-4051
Tian-Ming Fu  http://orcid.org/0000-0001-6265-0859
Liqun Luo  http://orcid.org/0000-0001-5467-9264

## Decision letter and Author response

Decision letter https://doi.org/10.7554/eLife.85521.sa1
Author response https://doi.org/10.7554/eLife.85521.sa2

# Additional files

## Supplementary files

• Supplementary file 1. Sample variability among individual brains. A supplemental table describing the biological and technical variations we observed among individual brain samples, and measures we took to minimize them, if possible.

• MDAR checklist

## Data availability

Figure 3—source data 1, Figure 5—source data 1, Figure 6—source data 1, and Figure 7—source data 1 contain the numerical and statistical data used to generate the figures. The confocal imaging dataset is available at Brain Image Library under DOI https://doi.org/10.35077/g.933.

The following dataset was generated:

| Author(s) | Year | Dataset title | Dataset URL | Database and Identifier |
|---|---|---|---|---|
| Wong KLK | 2023 | Origin of wiring specificity in an olfactory map revealed by neuron type-specific, time-lapse imaging of dendrite targeting: Confocal imaging of developing fly brain | https://doi.org/10.35077/g.933 | Brain Image Library, 10.35077/g.933 |

The following previously published dataset was used:

| Author(s) | Year | Dataset title | Dataset URL | Database and Identifier |
|---|---|---|---|---|
| Xie Q, Brbic M, Horns F, Kolluru SS, Jones RC, Li J, Reddy AR, Xie A, Kohani S, Li Z, McLaughlin CN, Li T, Xu C, Vacek D, Luginbuhl DJ, Leskovec J, Quake SR, Luo L Li H | 2021 | Temporal evolution of single-cell transcriptomes of *Drosophila* olfactory projection neurons | https://www.ncbi.nlm.nih.gov/geo/query/acc.cgi?acc=GSE161228 | NCBI Gene Expression Omnibus, GSE161228 |

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

# Appendix 1

## Appendix 1—key resources table

| Reagent type (species) or resource | Designation | Source or reference | Identifiers | Additional information |
|---|---|---|---|---|
| Genetic reagent (*D. melanogaster*) | *GH146-FLP* | DOI: 10.1038/nn.2442 | | |
| Genetic reagent (*D. melanogaster*) | *QUAS-FRT-stop-FRT-mCD8-GFP* | DOI: 10.1016 /j. cell.2010.02.025 | | |
| Genetic reagent (*D. melanogaster*) | *UAS-mCD8-GFP* | DOI: 10.1016 / s0896-6273(00)80701–1 | | |
| Genetic reagent (*D. melanogaster*) | *UAS-mCD8-FRT-GFP-FRT-RFP* | DOI: 10.1016 /j. neuron.2014.06.026 | | |
| Genetic reagent (*D. melanogaster*) | *VT033006-GAL4* | DOI: 10.1101/198648 | | |
| Genetic reagent (*D. melanogaster*) | *Mz19-GAL4* | DOI: 10.1242/dev.00896 | | |
| Genetic reagent (*D. melanogaster*) | *91 G04-GAL4* | DOI: 10.1016 /j. celrep.2012.09.011 | | |
| Genetic reagent (*D. melanogaster*) | *Mz612-GAL4* | DOI: 10.1242/dev.01614 | | |
| Genetic reagent (*D. melanogaster*) | *71B05-GAL4* | DOI: 10.1016 /j. celrep.2012.09.011 | | |
| Genetic reagent (*D. melanogaster*) | *Split7-GAL4* | DOI: 10.7554/eLife.63450 | | FlyLight:SS01867 |
| Genetic reagent (*D. melanogaster*) | *QUAS-FLP* | DOI: 10.1016 /j. cell.2010.02.025 | | |
| Genetic reagent (*D. melanogaster*) | *UAS-EcR.B1-ΔC655.F645A* | DOI: 10.1242/dev.00205 | | |
| Genetic reagent (*D. melanogaster*) | *tsh-GAL4* | Bloomington *Drosophila* Stock Center | BDSC:3040 | |
| Genetic reagent (*D. melanogaster*) | *lov-GAL4* | Bloomington *Drosophila* Stock Center | BDSC:3737 | |
| Genetic reagent (*D. melanogaster*) | *UAS-mCD8-GFP, hs-FLP; FRTG13, tub-GAL80;; GH146-GAL4* | DOI: 10.1016 / s0896-6273(00)80701–1 | | |
| Genetic reagent (*D. melanogaster*) | *FRTG13, UAS-mCD8-GFP* | DOI: 10.1016 / s0896-6273(00)80701–1 | | |
| Genetic reagent (*D. melanogaster*) | *UAS-FRT10-stop-FRT10-3xHalo7-CAAX* | this paper | | on either II or III chromosome; see **Materials and methods** |
| Genetic reagent (*D. melanogaster*) | *UAS-FRT-myr-4xSNAPf-FRT-3xHalo7-CAAX* | this paper | | on III chromosome; see **Materials and methods** |
| Genetic reagent (*D. melanogaster*) | *UAS-FRT-myr-mGreenLantern-FRT-3xHalo7-CAAX* | this paper | | on II chromosome; see **Materials and methods** |
| Genetic reagent (*D. melanogaster*) | *QUAS-FRT-stop-FRT-myr-4xSNAPf* | this paper | | on III chromosome; see **Materials and methods** |
| Genetic reagent (*D. melanogaster*) | *run-T2A-FLP* | this paper | | on X chromosome; see **Materials and methods** |
| Genetic reagent (*D. melanogaster*) | *acj6-T2A-FLP* | this paper | | on X chromosome; see **Materials and methods** |
| Genetic reagent (*D. melanogaster*) | *acj6-T2A-QF2* | this paper | | on X chromosome; see **Materials and methods** |
| Genetic reagent (*D. melanogaster*) | *CG14322-T2A-QF2* | this paper | | on III chromosome; see **Materials and methods** |
| Genetic reagent (*D. melanogaster*) | *lov-T2A-QF2* | this paper | | on II chromosome; see **Materials and methods** |
| Antibody | *chicken polyclonal anti-GFP* | Aves Lab | RRID:AB_10000240; Aves Lab:GFP-1020 | (1:1000) |

*Appendix 1 Continued on next page*

*Appendix 1 Continued*

| Reagent type (species) or resource | Designation | Source or reference | Identifiers | Additional information |
|---|---|---|---|---|
| Antibody | *rabbit polyclonal anti-DsRed* | TaKaRa | RRID:AB_10013483; TaKaRa:632496 | (1:500) |
| Antibody | *rat monoclonal anti-Cadherin DN* | Developmental Studies Hybridoma Bank | RRID:AB_528121; DSHB:DN-Ex#8 | (1:30) |
| Antibody | *mouse monoclonal anti-Bruchpilot* | Developmental Studies Hybridoma Bank | RRID:AB_2314866; DSHB:nc82 supernatant | (1:30) |
| Recombinant DNA reagent | *pBPGUw-HACK-QF2* | Addgene | RRID:Addgene_80276 | |
| Recombinant DNA reagent | *pU6-BbsI-chiRNA* | Addgene | RRID:Addgene_45946 | |
| Recombinant DNA reagent | *pUAS-3xHalo7-CAAX* | Addgene | RRID:Addgene_87646 | |
| Recombinant DNA reagent | *pUAS-myr-4xSNAPf* | Addgene | RRID:Addgene_87637 | |
| Recombinant DNA reagent | *pcDNA3.1-mGreenLantern* | Addgene | RRID:Addgene_161912 | |
| Recombinant DNA reagent | *p5XQUAS* | Addgene | RRID:Addgene_24349 | |
| Recombinant DNA reagent | *p10xQUAS-CsChrimson* | Addgene | RRID:Addgene_163629 | |
| Recombinant DNA reagent | *pUAS-FRT10-stop-FRT10-3xHalo7-CAAX* | this paper | | backbone from pUAS-3xHalo7-CAAX; see **Materials and methods** |
| Recombinant DNA reagent | *pUAS-FRT-myr-4xSNAPf-FRT-3xHalo7-CAAX* | this paper | | backbone from pUAS-3xHalo7-CAAX; see **Materials and methods** |
| Recombinant DNA reagent | *pUAS-FRT-myr-mGreenLantern-FRT-3xHalo7-CAAX* | this paper | | backbone from pUAS-3xHalo7-CAAX; see **Materials and methods** |
| Recombinant DNA reagent | *pUAS-myr-mGreenLantern* | this paper | | backbone from pUAS-myr-4xSNAPf; see **Materials and methods** |
| Recombinant DNA reagent | *pQUAS-FRT-stop-FRT-myr-4xSNAPf* | this paper | | backbone from p5XQUAS; see **Materials and methods** |
| Chemical compound, drug | SYLGARD 184 Silicone Elastomer Kit | DOW | DOW:2646340 | |
| Chemical compound, drug | Schneider's *Drosophila* Medium | ThermoFisher Scientific | ThermoFisher Scientific:21720001 | |
| Chemical compound, drug | Fetal Bovine Serum | ThermoFisher Scientific | ThermoFisher Scientific:16140071 | used at 10% |
| Chemical compound, drug | Human recombinant insulin | ThermoFisher Scientific | ThermoFisher Scientific:12585014 | used at 10 µg/mL |
| Chemical compound, drug | Penicillin-Streptomycin | ThermoFisher Scientific | ThermoFisher Scientific:15140122 | (1:100) |
| Chemical compound, drug | Ascorbic acid | Sigma | Sigma:A4544 | used at 50 mg/mL in water |
| Chemical compound, drug | 20-hydroxyecdysone | Sigma | Sigma:H5142 | used at 20 µM and 2 µM |
| Chemical compound, drug | JF503-cpSNAP | DOI: 10.1038/nmeth.4403; DOI: 10.1021/jacsau.1c00006 | | (1:1000); gift from Dr. Luke Lavis |
| Chemical compound, drug | JF646-Halo | DOI: 10.1038/nmeth.4403; DOI: 10.1021/jacsau.1c00006 | | (1:1000); gift from Dr. Luke Lavis |
| Chemical compound, drug | JFX650-SNAP | DOI: 10.1038/nmeth.4403; DOI: 10.1021/jacsau.1c00006 | | (1:1000); gift from Dr. Luke Lavis |
| Chemical compound, drug | JFX554-Halo | DOI: 10.1038/nmeth.4403; DOI: 10.1021/jacsau.1c00006 | | (1:10000); gift from Dr. Luke Lavis |

*Appendix 1 Continued on next page*

*Appendix 1 Continued*

| Reagent type (species) or resource | Designation | Source or reference | Identifiers | Additional information |
|---|---|---|---|---|
| Chemical compound, drug | JF635-Halo | DOI: 10.1038/nmeth.4403; DOI: 10.1021/jacsau.1c00006 | | (1:1000); gift from Dr. Luke Lavis |
| Chemical compound, drug | JF570-Halo | DOI: 10.1038/nmeth.4403; DOI: 10.1021/jacsau.1c00006 | | (1:5000); gift from Dr. Luke Lavis |
| Chemical compound, drug | Sulforhodamine 101 | Sigma | Sigma:S7635 | used at 1 µM |
| Software, algorithm | ZEN | Carl Zeiss | RRID:SCR_013672 | |
| Software, algorithm | ImageJ | National Institutes of Health | RRID:SCR_003070 | |
| Software, algorithm | Python Programming Language | Python | RRID:SCR_008394 | http://www.python.org/ |

