## [Editor Report]

When a neuron is born it correlates with where it targets in the neuropil and this has been best demonstrated in the olfactory lobe of *Drosophila*. This important study uses sophisticated genetics and advanced live imaging to provide a compelling description of how neuronal dendrites explore the target field, eliminate excessive branches, and assort into the correct region during development. In the process, it develops valuable tools. The study brings us closer to a comprehensive understanding of how the birth order of a neuron translates to dendrite patterning within the *Drosophila* antennal lobe circuit.

---

## [Decision Letter]

**Decision letter after peer review:**

Thank you for submitting your article "Origin of wiring specificity in an olfactory map: dendrite targeting of projection neurons" for consideration by *eLife*. Your article has been reviewed by 3 peer reviewers, including Sonia Sen as Reviewer #1 and Reviewing Editor, and the evaluation has been overseen by K VijayRaghavan as the Senior Editor.

Essential revisions:

1. Quantifications: While the data are qualitatively convincing, could the authors please quantify their observations and mention the sample size for each of their experiments? It would be useful to have a way of describing the variability between brains.

2. The time windows: could the authors please define their time windows better, particularly in the context of other neurons that are born? (Please see detailed comments below)

3. Writing: Could the authors please place their work in the broader context of the literature on how temporal patterning translates to dendritic patterning? While doing so, could they also place it in the framework of formal possibilities of how this might occur?

4. Since the antennal lobe is a 3D structre, we would like the authors represent their 2D models as 3D? (Does their model hold up?)

We also suggest that the authors experimentally test their prediction. This could be by showing that a later-born neuron from adPN lineage will always target further clockwise. Or they could use Chinmo to change the temporal identity of the neurons. (Or any similar experiment they choose). If not, we suggest that they revisit their text to tone down their claim of dendrites targeting according to birth order.

Aside from these essential revisions, please read the detailed reviews below and address them whenever possible.

*Reviewer #1 (Recommendations for the authors):*

I appreciate the quality and extent of the work presented in this manuscript and have only comments related to its presentation.

1. I find great value in describing how the earliest events in the birth of the neuron – through temporal patterning – translate to its target specification. This is an important area in neurodevelopment and one into which inroads have been made in the recent past. Much of this work has been in the ventral nerve cord and the optic lobe. It would benefit this manuscript immensely to place their work within this context.

2. The manner in which the study is currently presented gives the (false) impression that this is 'merely' a descriptive study that conveys in more detail an already known phenomenon. This is likely because the authors have restricted their reference to literature on the antennal lobe. The authors should present this work in the framework of formal possibilities of how birth order might affect targeting. Currently, the articulation of the problem they are addressing is too generic.

*Reviewer #2 (Recommendations for the authors):*

Wong et al. used very sophisticated genetics to perform a thorough investigation of adPN and lPN lineages across various developmental stages to find out how dendrites segregate into discrete glomeruli during development. The endeavours devoted to data collection are very impressive. The data are convincing and adequately interpreted.

The studies are thorough. With the limitation of genetic tools available, Wong et al. are still able to achieve a comprehensive study of two neuronal lineages to extract wiring logic.

Ex vivo explant time-lapse imaging provides details of how dendritic neurites behave during development, which could not be reached with conventional standard fixation-staining protocols. I agreed with their words that '…the (power and) necessity of type-specific neuronal access and time-lase imaging in identifying wiring mechanisms…'. The methodology will become a paradigm in the field.

I have some specific comments for the authors:

1. Adult-specific antennal lobes were first built outside the larval antennal lobe and then took over the larval antennal lobe territory. They also observed that embryonic-born PNs retracted and extended dendrites simultaneously at spatially distinct regions. It was unexplained and undiscussed if the larval antennal lobes had a footprint left at that region to influence larval-born PN dendritic targeting, such as glial cells, unpruned dendrites, or presynaptic partners. And it remains to be addressed whether the rotation of initial dendritic targeting of PNs is related to the remnants of larval-specific antennal lobes.

2. Unlike the anterodorsal lineage that generates monoglomerulous PNs continuously, the lateral lineage intermingles the birth of monoglomerulous PNs (studied in this manuscript) and antennal mechanosensory and motor center (AMMC) PNs during neurogenesis (Lin et al., 2012, PLoS Biology). While it is convincing adPNs born around the same time form a cohort to target dendrites to similar territory, it is unclear if those AMMC PNs which are born between monoglomerulous PNs form a cohort with those PNs and target their dendrites to similar territory. Besides, more lPNs (7) than adPNs (3) were categorized as 'early-born', and I was wondering if those 'early-born' lPNs can be further partitioned into smaller cohorts if AMMC PNs are considered.

3. The authors proposed a compelling model about how PNs chose initial dendritic targeting territory based on its lineage and birth timing and showed a clock-like rotation as the targeting pattern. However, the antennal lobe is actually a 3D structure. 2D projection looks very intriguing, but it does not really reflect how targeting territory is selected in a 3D space. It might be more accurate to revise it as a 3D model.

4. The definition of 'early', 'middle', and 'late' born PNs from both lineages is not very clear. Based on birth timing (what interval?) or dendrite targeting? Why anterodorsal lineage has fewer early-born PNs than the lateral lineage (3 vs. 7)?

5. In this manuscript, many neurons from the lateral lineage were left undiscussed. It might be good to remind readers that this manuscript is focusing on the monoglomerulous PNs only; interneurons and other types of PNs from the lateral lineage won't be discussed in the current work.

*Reviewer #3 (Recommendations for the authors):*

I find the paper a strong candidate for *eLife*. The genetics is exceptional and the effort to comprehensively dissect the targeting of as many projection neurons as possible is both impressive and commendable. The ex-vivo time-lapse imaging is likewise super-impressive.

A few points that should be strengthened in my opinion:

1) Most of the figures represent convincing qualitative information without giving us any quantitative measures. At a very minimum, I would like to know how many neurons or brains (or both) were images and gave a consistent finding. Even better would be to find ways to describe the variability between brains. I understand that the lack of a "standard brain" for these developmental stages makes this more complicated but perhaps a course resolution would do.

2) I am not convinced by the statement that "groups" of neurons (grouped by birth date) target their dendrites to the same location. There are even a few projection neurons that belong to two groups – what does this mean? The clockwise rotation model should, in principle, offer a way to test this – no? It predicts that regardless of the stage, always the next-born neuron should target the clockwise correct location compared to the previous. This should strengthen the statement and verify if indeed there are groups – of indistinguishable (at this stage) dendrites…

3) While I don't necessarily think that every paper needs to include mechanistic experiments, the two-step model presented in this manuscript, coupled with the many dendrite-targeting mutants that the Luo lab has previously generated, really makes it compelling to check if they are required for the initial targeting or later refinement. So for example, is the initial age and lineage dependent targeting normal in Sema1a; Ten-a/m; etc… Of course – does the final targeting depend on correct initial targeting. This is not absolutely necessary but adding some mechanistic aspects would make the study much more compelling to me.

4) If I understand correctly, previous data from the Luo lab has shown that the ORN-PN map is extremely stable, even if you kill or inactivate specific ORN/PNs. In light of this new study, PN-PN interactions have the potential to be important. Could adPNs affect the targeting of lPNs? Or vice versa? If this was not tested before, then perhaps even just discussing the option, rather than doing the experiments, seems like a logical step to me.

---

## [Author Response]

Essential revisions:1. Quantifications: While the data are qualitatively convincing, could the authors please quantify their observations and mention the sample size for each of their experiments? It would be useful to have a way of describing the variability between brains.

Thank you for pointing out the need of quantification. Sample size for each experiment in Figures 6 and 7 has already been provided during the initial submission of the manuscript. We now include the sample size for experiments in Figures 1, 3, 4, 5 and 8. This information can be found in the figure legends.

We reason that the variability among sample brains could arise from biological (e.g., developmental rates, cell number of each neuronal type, and cell body positions) and technical variations (e.g., brain mounting, staining efficiency, genetic design). We now include a supplemental table describing the variations we observed, and what measures we have taken to minimize them, if possible (Supplementary file 1).

We had once considered providing additional samples for each genotype (just like what we did for the MARCM experiments; Figure 3D_4–9_, Figure 3 —figure supplement 3B–E, Figure 4B). However, we observed very stereotyped dendrite targeting of a given PN across development (see DL1 PNs as an example in Figure 3A_3_, 3D_1–3_), and presentation of all data comprising various genotypes and developmental stages would be overwhelming for readers. In the main figures, we present the most representative images selected from reproducible dataset so that readers can focus on the wiring logic that organizes different types of PNs into a neural circuit. To help researchers interact directly with our imaging data, we are in the process of depositing raw confocal images into Brain Image Library (https://www.brainimagelibrary.org/). Relevant links will be provided when available. This should allow one to examine the dendrite patterning of specific PN types at developmental stages of interest, stack-by-stack or at any angle, and to examine the variability among individual samples.

2. The time windows: could the authors please define their time windows better, particularly in the context of other neurons that are born? (Please see detailed comments below)

We now add text to describe how we define the approximate temporal cohorts of PNs and include the time intervals during which we applied the 1-hour heat shock to generate single-cell MARCM clones. Please see details in our response to Comment #4 from Reviewer #2.

3. Writing: Could the authors please place their work in the broader context of the literature on how temporal patterning translates to dendritic patterning? While doing so, could they also place it in the framework of formal possibilities of how this might occur?

We totally agree with Editors and Reviewers that there is a rich literature on the temporal patterning of the ventral nerve cord and optic lobe neuroblasts, and the discoveries of the temporal transcription factor cascades are remarkable (e.g., Doe, 2017; Miyares and Lee, 2019). In the antennal lobe, Chinmo, a temporal transcription factor, and the temporal gradients of RNAbinding proteins that regulate Chinmo translation, have been shown to govern adPN cell fate (Zhu et al., 2006; Liu et al., 2015). From these studies, it is tempting to speculate that the approximate temporal cohorts of a given PN lineage could be the result of differential expression of temporal factors. Future studies investigating the molecular signatures of these cohorts should inform us how PNs of a given lineage translate birth order into dendrite patterning.

We have placed our work in broader context and discussed the potential molecular mechanisms based on previous literatures. Please see pages 12–13 – Lines 535–557.

We apologize for this oversight in our original submission, and sincerely thank *eLife* Editors and Reviewers for this critique.

4. Since the antennal lobe is a 3D structre, we would like the authors represent their 2D models as 3D? (Does their model hold up?)

We agree that the antennal lobe is a 3D structure despite its relatively short anterior-posterior axis at early stages (~20 µm at 12h APF). To visualize PN dendrite targeting in 3D, we generate videos showing 3D rendering of *z* stacks of labeled PN dendrites in 12h APF antennal lobes with rotation along the *y* axis (Figure 3 – video 1, Figure 4 – video 1 and Figure 8 – video 1).

3D visualization reveals that PN dendrites were “located primarily on the periphery of the antennal lobe, whereas the center housed the axon bundle projecting out of the antennal lobe. Some dendrites could reach almost the entire depth, suggesting active exploration of the surroundings in many directions. While 3D projections provide rich details in depth and different viewing angles, we did not find apparent relationship between birth order and dendrite targeting along the anterior-posterior axis, at least for the examined PN types at 12h APF. Thus, the approximate 2D projection (Figure 3E_2–4_) conveys the logic of dendrite patterning effectively.” Please see page 6 – Lines 240–249.

We also suggest that the authors experimentally test their prediction. This could be by showing that a later-born neuron from adPN lineage will always target further clockwise. Or they could use Chinmo to change the temporal identity of the neurons. (Or any similar experiment they choose). If not, we suggest that they revisit their text to tone down their claim of dendrites targeting according to birth order.

Using MARCM and specific driver lines, we have access to four approximate temporal cohorts of adPNs based on their birth timing: (1) early, (2) mid-early, (3) mid-late and (4) late larval born adPNs. Analyses of their dendrite targeting have shown that a later-born adPN will always target further clockwise (see Figure 3D).

We are also keen on understanding the molecular mechanisms linking PN temporal identity to initial dendrite targeting. Indeed, Chinmo is known to specify the temporal identity of adPNs (Chinmo protein level is high in early-born and low in late-born PNs); loss of *chinmo* in the firstborn DL1 PNs leads to mistargeting to glomerulus targeted by the fourth-born D PNs (Zhu *et al.*, 2006). This provides a molecular link between temporal identity and final glomerular targeting. However, as DL1 PNs and D PNs are both early-born PNs, we expect the changes in the initial dendrite targeting, if any, in DL1 PNs mutant for *chinmo* to be very subtle to observe. Currently, we are in the process of examining the transcriptome profiles among different cohorts to identify key molecules that create the rotation pattern. This will take many months of further work.

In the manuscript, we have avoided using strong statements like ‘birth order instructs dendrite targeting’.

Reviewer #1 (Recommendations for the authors):I appreciate the quality and extent of the work presented in this manuscript and have only comments related to its presentation.1. I find great value in describing how the earliest events in the birth of the neuron – through temporal patterning – translate to its target specification. This is an important area in neurodevelopment and one into which inroads have been made in the recent past. Much of this work has been in the ventral nerve cord and the optic lobe. It would benefit this manuscript immensely to place their work within this context.

We thank Reviewer #1 for pointing out the outstanding work investigating the temporal patterning in the ventral nerve cord and the optic lobe. We have added text and relevant references to the Discussion to place our study in this context. Please see our response to Essential Revision #3 for details.

2. The manner in which the study is currently presented gives the (false) impression that this is 'merely' a descriptive study that conveys in more detail an already known phenomenon. This is likely because the authors have restricted their reference to literature on the antennal lobe. The authors should present this work in the framework of formal possibilities of how birth order might affect targeting. Currently, the articulation of the problem they are addressing is too generic.

We took Reviewer #1’s advice and now discuss the possible molecular mechanisms underlying how PN birth order might affect dendrite targeting. Please see our response to Essential Revision #3 for details.

Reviewer #2 (Recommendations for the authors):I have some specific comments for the authors:1. Adult-specific antennal lobes were first built outside the larval antennal lobe and then took over the larval antennal lobe territory. They also observed that embryonic-born PNs retracted and extended dendrites simultaneously at spatially distinct regions. It was unexplained and undiscussed if the larval antennal lobes had a footprint left at that region to influence larval-born PN dendritic targeting, such as glial cells, unpruned dendrites, or presynaptic partners. And it remains to be addressed whether the rotation of initial dendritic targeting of PNs is related to the remnants of larval-specific antennal lobes.

We thank Reviewer #2 for pointing out the potential involvement of the larval-specific antennal lobe in the initial dendrite map formation of the adult-specific antennal lobe. We have added into Discussion the following quoted test from our lab found that “the larval-specific ORN axons secrete semaphorins, Sema-2a and Sema-2b, which act as repulsive ligands for dendrites of Sema-1a-expressing PNs (including DL1 PNs) (Komiyama *et al.*, 2007; Sweeney *et al.*, 2011).

As the larval-specific lobe is located ventromedial to the adult-specific lobe, Sema-2a/b and Sema-1a form opposing gradients along the dorsolateral-ventromedial axis. When DL1 PNs (the first-born/developed) begin to target their dendrites, this repulsive action could destabilize branches in the ventromedial direction and thus favor dorsolateral targeting. This provides a plausible explanation as to why the adPN rotation pattern begins at the dorsolateral position.” Please see page 13 – Lines 566–576.

2. Unlike the anterodorsal lineage that generates monoglomerulous PNs continuously, the lateral lineage intermingles the birth of monoglomerulous PNs (studied in this manuscript) and antennal mechanosensory and motor center (AMMC) PNs during neurogenesis (Lin et al., 2012, PLoS Biology). While it is convincing adPNs born around the same time form a cohort to target dendrites to similar territory, it is unclear if those AMMC PNs which are born between monoglomerulous PNs form a cohort with those PNs and target their dendrites to similar territory.

Reviewer #2 is correct that the lateral lineage produces 5 distinct PN classes in an intercalated manner: monoglomerulous PNs (mPNs), unilateral PNs, bilateral PNs, AMMC PNs and SOG PNs (Lin *et al.*, 2012). As only mPNs are *GH146*+, we have not characterized the patterning of the other 4 PN types and therefore do not know if AMMC PNs born between mPNs share a similar targeting territory at any developmental stage.

Despite the lack of data, we find the point raised by Reviewer #2 very intriguing. Although AMMC PNs do not innervate the antennal lobe in the adult brain, whether they do so during development is not known. Cell-type specific labeling in the early developing brain, similar to what we have done for the mPNs, could provide invaluable insights into how neurons produced from the same lineage contribute to multiple circuitries that endow diverse sensory modalities. Nonetheless, as we focus on how wiring specificity arises in the olfactory map, analyses of other types of PNs are beyond the scope of the current work.

Please find relevant text changes in the Introduction: page 2 – Lines 79–82.

Besides, more lPNs (7) than adPNs (3) were categorized as 'early-born', and I was wondering if those 'early-born' lPNs can be further partitioned into smaller cohorts if AMMC PNs are considered.

Unfortunately, we currently do not have tools to label early-born lPNs at higher resolution, and therefore do not know whether the early-born lPNs can furthered be partitioned into smaller cohorts. Neither do we know whether AMMC PNs born in between would act as separators.

3. The authors proposed a compelling model about how PNs chose initial dendritic targeting territory based on its lineage and birth timing and showed a clock-like rotation as the targeting pattern. However, the antennal lobe is actually a 3D structure. 2D projection looks very intriguing, but it does not really reflect how targeting territory is selected in a 3D space. It might be more accurate to revise it as a 3D model.

We totally agree that a 3D model would provide a clearer picture of dendrite targeting during the initial map formation, and therefore provide videos showing the 3D visualization. Please see our detailed response in Essential Revision #4.

4. The definition of 'early', 'middle', and 'late' born PNs from both lineages is not very clear. Based on birth timing (what interval?) or dendrite targeting? Why anterodorsal lineage has fewer early-born PNs than the lateral lineage (3 vs. 7)?

We note that the heat shock time window to induce MARCM clones of the first-born DL1 PNs is wide (from 0 to 60h ALH), as reported previously (Jefferis *et al.*, 2001). When we applied heat shock at 42–48h ALH to access early-born adPNs, most of the clones (>80%) were still DL1 PNs (Figure 3 —figure supplement 2E). This is likely because the neuroblast that gives rise to adPNs is arrested at G2 until quite sometime after larval hatching. This might cause the difference in the number of PN types in the early-born cohorts between adPN and lPN lineages.

5. In this manuscript, many neurons from the lateral lineage were left undiscussed. It might be good to remind readers that this manuscript is focusing on the monoglomerulous PNs only; interneurons and other types of PNs from the lateral lineage won't be discussed in the current work.

We thank Reviewer #2 for the advice and now remind readers that our work focuses on the wiring specificity of monoglomerular PNs only. This has been added to the Introduction. Please see page 2 – Lines 79–82.

Reviewer #3 (Recommendations for the authors):I find the paper a strong candidate for eLife. The genetics is exceptional and the effort to comprehensively dissect the targeting of as many projection neurons as possible is both impressive and commendable. The ex-vivo time-lapse imaging is likewise super-impressive.A few points that should be strengthened in my opinion:1) Most of the figures represent convincing qualitative information without giving us any quantitative measures. At a very minimum, I would like to know how many neurons or brains (or both) were images and gave a consistent finding. Even better would be to find ways to describe the variability between brains. I understand that the lack of a "standard brain" for these developmental stages makes this more complicated but perhaps a course resolution would do.

We thank Reviewer #3 for pointing out the need of quantifications. We now provide the sample size for each genotype in the figure legends. We also add a new table (Supplementary file 1) describing different types of variations among individuals as well as measures we took to minimize them. Please see details in our response to Essential Revision #1.

2) I am not convinced by the statement that "groups" of neurons (grouped by birth date) target their dendrites to the same location. There are even a few projection neurons that belong to two groups – what does this mean? The clockwise rotation model should, in principle, offer a way to test this – no? It predicts that regardless of the stage, always the next-born neuron should target the clockwise correct location compared to the previous. This should strengthen the statement and verify if indeed there are groups – of indistinguishable (at this stage) dendrites…

We apologize for the use of “groups”, which might falsely imply that PNs themselves are intrinsically arranged into discrete groups based on their birth order. We have now used “approximate temporal cohorts” instead of “groups” and defined the cohorts “based on birth timing that corresponds to the heat shock time we applied to induce single-cell MARCM clones”. Please see detailed definition of each cohort in page 5 – Lines 209–215.

Indeed, “we note that DM6 and VA1v PNs were assigned to both cohorts of mid-late and lateborn adPNs, reflecting the nature of short birth timing differences and overlaps between adjacent cohorts” (page 2 – Lines 215–217). This also suggests that PNs are unlikely to be arranged into discrete groups.

For experiments to test the rotation model, please see our response in Essential Revision.

We have removed phrases such as “grouping by birth order” from the text and figures.

3) While I don't necessarily think that every paper needs to include mechanistic experiments, the two-step model presented in this manuscript, coupled with the many dendrite-targeting mutants that the Luo lab has previously generated, really makes it compelling to check if they are required for the initial targeting or later refinement. So for example, is the initial age and lineage dependent targeting normal in Sema1a; Ten-a/m; etc… Of course – does the final targeting depend on correct initial targeting. This is not absolutely necessary but adding some mechanistic aspects would make the study much more compelling to me.

We thank Reviewer #3 for raising these questions. Previous studies from our lab indeed demonstrate the loss of Sema-1a causes mistargeting of DL1 PN dendrites as early as 16h APF.

More importantly, such mistargeting phenotypes were observed consistently throughout development as well as in adulthood. These pieces of evidence illustrate that the initial dendrite targeting is important to the final targeting. We now add text describing the importance of the initial map in the Discussion. Please see page 14 – Lines 601–608.

4) If I understand correctly, previous data from the Luo lab has shown that the ORN-PN map is extremely stable, even if you kill or inactivate specific ORN/PNs. In light of this new study, PN-PN interactions have the potential to be important. Could adPNs affect the targeting of lPNs? Or vice versa? If this was not tested before, then perhaps even just discussing the option, rather than doing the experiments, seems like a logical step to me.

Reviewer #3 is right about the wiring stability of the adult olfactory map once PN-ORN connections are established (Berdnik *et al.*, 2006). We note that because of the technical limitations (availability of drivers with early onset), the Berdnik et al. study was restricted to perturbing the olfactory circuit after wiring specificity has largely been established.

Given the robust PN dendritic dynamics seen in initial targeting process (Figures 5–8), we agree with the reviewer that whether adPNs and lPNs may reciprocally affect dendrite targeting is a very intriguing question. Although we currently do not have data to provide answers, we discuss the experimental designs that could address it in future works. See page 13 – Lines 558–565 in the Discussion.